# A spin-refrigerated cavity quantum electrodynamic sensor

Hanfeng Wang[1], Kunal L. Tiwari[2], Kurt Jacobs [3,4], Michael Judy[5], Xin Zhang[5], Dirk R. Englund [1]✉ & Matthew E. Trusheim [1,3]✉

Quantum sensors based on solid-state defects, in particular nitrogen-vacancy (NV) centers in diamond, enable precise measurement of magnetic fields, temperature, rotation, and electric fields. Cavity quantum electrodynamic (cQED) readout, in which an NV ensemble is hybridized with a microwave mode, can overcome limitations in optical spin detection and has resulted in leading magnetic sensitivities at the pT-level. This approach, however, remains far from the intrinsic spin-projection noise limit due to thermal Johnson-Nyquist noise and spin saturation effects. Here we tackle these challenges by combining recently demonstrated spin refrigeration techniques with comprehensive nonlinear modeling of the cQED sensor operation. We demonstrate that the optically-polarized NV ensemble simultaneously provides magnetic sensitivity and acts as a heat sink for the deleterious thermal microwave noise background, even when actively probed by a microwave field. Optimizing the NV-cQED system, we demonstrate a broadband sensitivity of $576 \pm 6 \, \text{fT}/\sqrt{\text{Hz}}$ around 15 kHz in ambient conditions. We then discuss the implications of this approach for the design of future magnetometers, including near-projection-limited devices approaching $3 \, \text{fT}/\sqrt{\text{Hz}}$ sensitivity enabled by spin refrigeration.

Quantum sensors offer the prospect of device operation at the physical limit of performance[1]. Among the many quantum sensing systems, nitrogen-vacancy (NV) centers in diamond have emerged as a promising platform[2–5] due to favorable attributes including room-temperature spin polarization and readout, atomic-scale size[6–9], and long coherence times[10,11]. NV-based sensors use resonance spectroscopy of the ground-state spin triplet transition frequencies to infer environmental properties, yielding excellent performance in a variety of sensing modalities including magnetometry[12–14], electrometry[15–17] and inertial sensing[18,19], even in extreme regimes[20]. Compared with other sensitive magnetometers such as superconducting quantum interference devices or spin exchange relaxation-free atomic vapors[21–23], NVs can operate in ambient conditions without cryogenics or magnetic shielding, but their sensitivity is limited to pT-level[2,13,24,25]. For spin magnetometers, the standard quantum limit (SQL) of

sensitivity is $\eta = 1/\gamma_e \sqrt{N T_2^*}$, where $\gamma_e$ is the gyromagnetic ratio, $T_2^*$ is the spin dephasing time, and $N$ is the number of spins. There is a tradeoff between $N$ and the ability to measure the spins, which is quantified as the inverse readout fidelity $\sigma_e$ – the ratio between achieved sensitivity and the SQL[2]. For a single-NV sensor, $\sigma_e \sim 1$ can be achieved but the sensitivity is limited to the nT-level[26]. NV ensemble sensors using continuous optical approaches have significantly increased $\sigma_e$ to $\sim 5000$, but the enhancement from the increased spin number $\sqrt{N} \sim 10^6$ leads to a pT-level sensitivity[2,25,27]. Approaching the SQL while maintaining spin ensemble size offers the potential for orders of magnitude in performance improvement.

To overcome the limitations of optical readout, the NV spin resonance may be probed in the microwave domain by coupling the NV ensemble to a microwave cavity mode. This approach eliminates fluorescence shot noise as the dominant systematic noise. The-state-

[1]Massachusetts Institute of Technology, Cambridge, MA, USA. [2]MIT Lincoln Laboratory, Lexington, MA, USA. [3]DEVCOM Army Research Laboratory, Adelphi, MD, USA. [4]Department of Physics, University of Massachusetts Boston, Boston, MA, USA. [5]Analog Devices, Inc., Wilmington, MA, USA. ✉e-mail: englund@mit.edu; matthew.e.trusheim.civ@army.mil

of-the-art[24] reported sensitivity for this approach is 3 pT/$\sqrt{\text{Hz}}$ but remains a factor of $\sigma_e \sim 1800$ above the SQL. The sensitivity of this approach is limited by the signal-to-noise ratio of the microwave readout, where the signal is upper-bounded by spin saturation effects, and the noise floor is determined by thermal Johnson-Nyquist noise. In this work, we address both of these limitations. We comprehensively model nonlinear spin saturation effects in the strong coupling regime, including spin ensemble inhomogeneity and optical polarization dynamics. We further apply recently introduced "spin refrigeration"[28–31] techniques to an actively-probed sensor for the first time, reducing the noise floor below the Johnson-Nyquist limit. We combine these two effects in the nonlinear regime to optimize device performance, achieving a sensitivity of 576 ± 6 fT/$\sqrt{\text{Hz}}$ with an inverse readout fidelity of $\sigma_e \sim 360$ which sets the state-of-the-art for a continuous-wave NV magnetometer.

Beyond the present experimental results, our model allows us to outline sensitivity regimes for varying spin ensemble properties, cavity designs, and power requirements. We argue for the feasibility of 3 fT/$\sqrt{\text{Hz}}$ sensitivity in an optimized device with a $\sigma_e \sim 33$. The demonstration of spin refrigeration in an operating sensor shows the potential for improvement of any Johnson-Nyquist limited system, e.g. micro-electromechanical systems.

## Results

### cQED Magnetometer Optimization

We seek to optimize the sensitivity of a cavity quantum electrodynamic (cQED) hybrid system consisting of an NV-doped diamond within a high-quality-factor ($Q$) dielectric resonator, shown schematically in Fig. 1a, b. We probe the cQED device with an external microwave field and measure the reflected microwave signal in the presence of varying magnetic fields. The sensitivity of the cQED device to magnetic fields can be expressed as

$$\eta = \frac{\sqrt{3}}{\gamma_e} \frac{\sqrt{\mathcal{L}}}{S},$$ (1)

for a low-frequency external stimulus and large post-reflection gain. Here $\mathcal{L}$ is the noise power spectral density, and $S$ is the change in detected signal per unit of NV frequency shift. We consider the magnetic field along [100] for equal response from each NV orientation, which leads to a sensitivity increase by a factor of $\sqrt{3}$. To optimize $\eta$ we break the problem into two parts: maximization of $S$ and minimization of $\mathcal{L}$.

### Sensor response

To reach optimal $S$, a natural approach is to maximize microwave probe power and therefore power response to any reflectivity change. In this regime, however, the strong microwave drive broadens the NV lines, reduces spin polarization, and the reflection coefficient acquires a dependence on probe power. While the nonlinear spectroscopic properties of related homogeneous systems have been established[32], the ensemble inhomogeneous distribution plays a central role in determining the reflection coefficient in this strong-driving regime. Previous cavity-coupled sensor reports have noted the importance of ensemble inhomogeneity in determining performance, and have treated this problem phenomenologically[24,33]. Related work describing bistability and critical dynamical behavior for transiently-driven inhomogeneous systems[34] are also well described by numerical treatments[35,36]. Here we extend those works in the context of sensing, determining the parametric dependence of the system response and saturation threshold on the inhomogeneous linewidth at the frequency tuning relevant to device operation.

We model the $N \gg 1$ NV centers as non-interacting two-level systems with transition frequencies $\omega_j$, distributed inhomogeneously due to heterogeneous local magnetic and strain environments as well as hyperfine coupling with $^{14}$N nuclear spins (Fig. 1b). The center frequency of the spin transitions, $\omega_s$, is tuned by modulating the amplitude of a uniform magnetic field along the diamond's [100] axis. The spin-selective optical polarization cycle drives the spins towards polarization $\bar{\mathcal{P}}$ with polarization rate $\gamma_p$. The homogeneous linewidth $\gamma$ is modeled to have an intrinsic value $\gamma_0$ as well as an additive optical-excitation-rate-dependent term (see Supplementary Note IID). The cavity mode has loaded relaxation rate $\kappa = \kappa_c + \kappa_{c1}$ for intrinsic relaxation rate $\kappa_c$ and coupling strength $\kappa_{c1}$ to a microwave probe line. Finally, we assume uniform coupling strength $g_s$ between each individual spin and the cavity mode.

Under an input microwave drive field $\beta_{in}$, the operator expectation values of the cavity field $\alpha$, spin coherence $s_j$, and polarization $\mathcal{P}_j$ (defined as the population difference between the $m_s = 0$ and $m_s = 1$ populations) obey semi-classical equations of motion:[37–39]

$$\dot{\alpha} = -\left(i\Delta + \frac{\kappa}{2}\right)\alpha - ig_s \sum_j s_j + \sqrt{\kappa_{c1}}\beta_{in}$$

$$\dot{s}_j = -\left(i\Delta_j + \frac{\gamma}{2}\right)s_j - ig_s \mathcal{P}_j \alpha$$ (2)

$$\dot{\mathcal{P}}_j = -\gamma_p\left(\mathcal{P}_j - \bar{\mathcal{P}}\right) - 2ig_s\left(s_j \alpha^* - s_j^* \alpha\right).$$

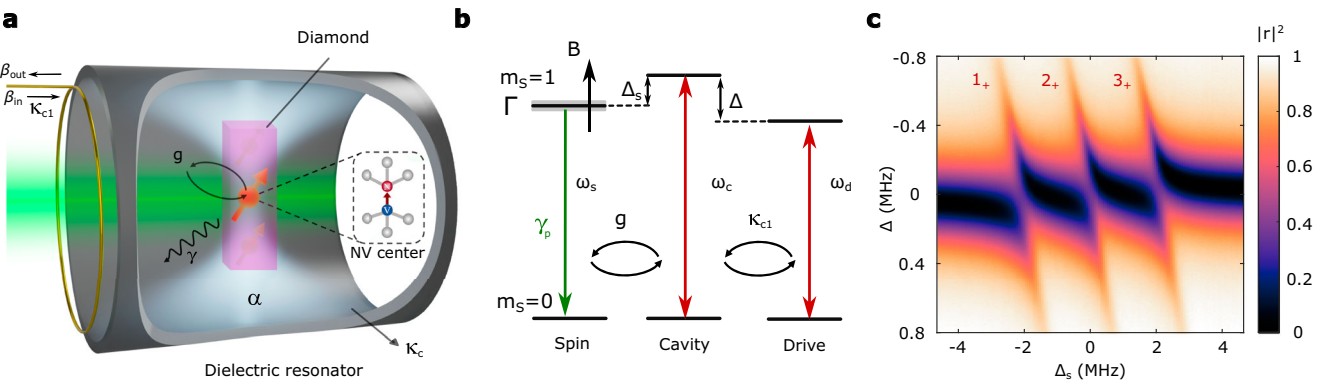

**Fig. 1 | Hybrid NV-cavity system. a** A diamond containing an NV ensemble is located at the mode maximum of the TE01$\delta$ mode of a dielectric resonator. A green laser is applied to continuously polarize the NV spins to the $|m_s = 0\rangle$ electronic ground state, and a detection loop is incorporated to electrically probe the system. The cavity field is denoted as $\alpha$, the input drive field as $\beta_{in}$, detected output field as $\beta_{out}$, the NV-cavity coupling rate $g$, NV decoherence rate $\gamma$, cavity intrinsic loss rate as $\kappa_c$ and cavity-drive coupling $\kappa_{c1}$. **b** NV-cavity energy level structure. The NV electron spin $m_s = 0$ to $m_s = 1$ transition, cavity, and drive fields have frequencies $\omega_s$, $\omega_c$, and $\omega_d$ respectively, resulting in spin-cavity detuning $\Delta_s = \omega_c - \omega_s$ and drive-cavity detuning $\Delta = \omega_c - \omega_d$. The NV spin transition frequency is tuned with an external magnetic field $B$, has an inhomogeneous width $\Gamma$, and is optically polarized at a rate $\gamma_p$. **c** Reflection $|r|^2$ as a function of spin-cavity and drive-cavity detuning. Three avoided crossings, labeled $1_+$, $2_+$, and $3_+$, indicate strong coupling to each of the three NV sub-ensembles shifted by $^{14}$N hyperfine coupling.

Here, $\Delta_j = \omega_j - \omega_d$ is the detuning between the NV transition frequency $\omega_j$ and the drive frequency $\omega_d$, and $\Delta = \omega_c - \omega_d$ is the detuning between the cavity frequency $\omega_c$ and the drive frequency. The reflected signal $\beta_{out}$ obeys the input-output relation $\beta_{out} = \sqrt{\kappa_{c1}}\alpha - \beta_{in}$. The response of $r = \beta_{out}/\beta_{in}$ to environmental changes which modify the spin transition frequencies allows the device to operate as a sensor. This can be seen in Fig. 1c, where tuning of the spin frequency $\Delta_s$ across the cavity results in a clear change in reflection coefficient $|r|^2$. We denote the constant of proportionality between the reflected voltage amplitude and the NV ensemble frequency shift as the signal $S$:

$$S(\beta_{in}) = \left| f(\beta_{in}) \frac{\partial \text{Im}[r]}{\partial \omega_s} \right|. \tag{3}$$

Here $f(\beta_{in})$ is the input probe voltage and is proportional to the input field $\beta_{in}$.

As $f(\beta_{in})$ has monotonic, linear dependence with $\beta_{in}$, increased probe power will result in increased signal for fixed reflectivity change $\partial \text{Im}[r]/\partial \omega_s$. However, in the limit of large drive power on resonance with both the cavity and the spin ensemble, the spin polarization is suppressed and the ensemble is effectively decoupled from the resonator. This results in a nonlinearity: the reflection coefficient itself changes with input field, meaning $\beta_{out}$ is no longer linearly related to $\beta_{in}$. This effect is shown in Fig. 2a, with full spectra at selected powers shown in Fig. 2b.

We determine the onset of this nonlinear saturation behavior by examining the conditions for which the cavity occupancy, $|\alpha|^2$ is no longer linearly proportional to the microwave drive power. The steady-state occupancy of the cavity obeys the nonlinear equation

$$|\beta_{in}|^2 = \frac{\kappa}{4} \frac{\kappa}{\kappa_{C1}} (1 + C_\alpha)^2 |\alpha|^2, \tag{4}$$

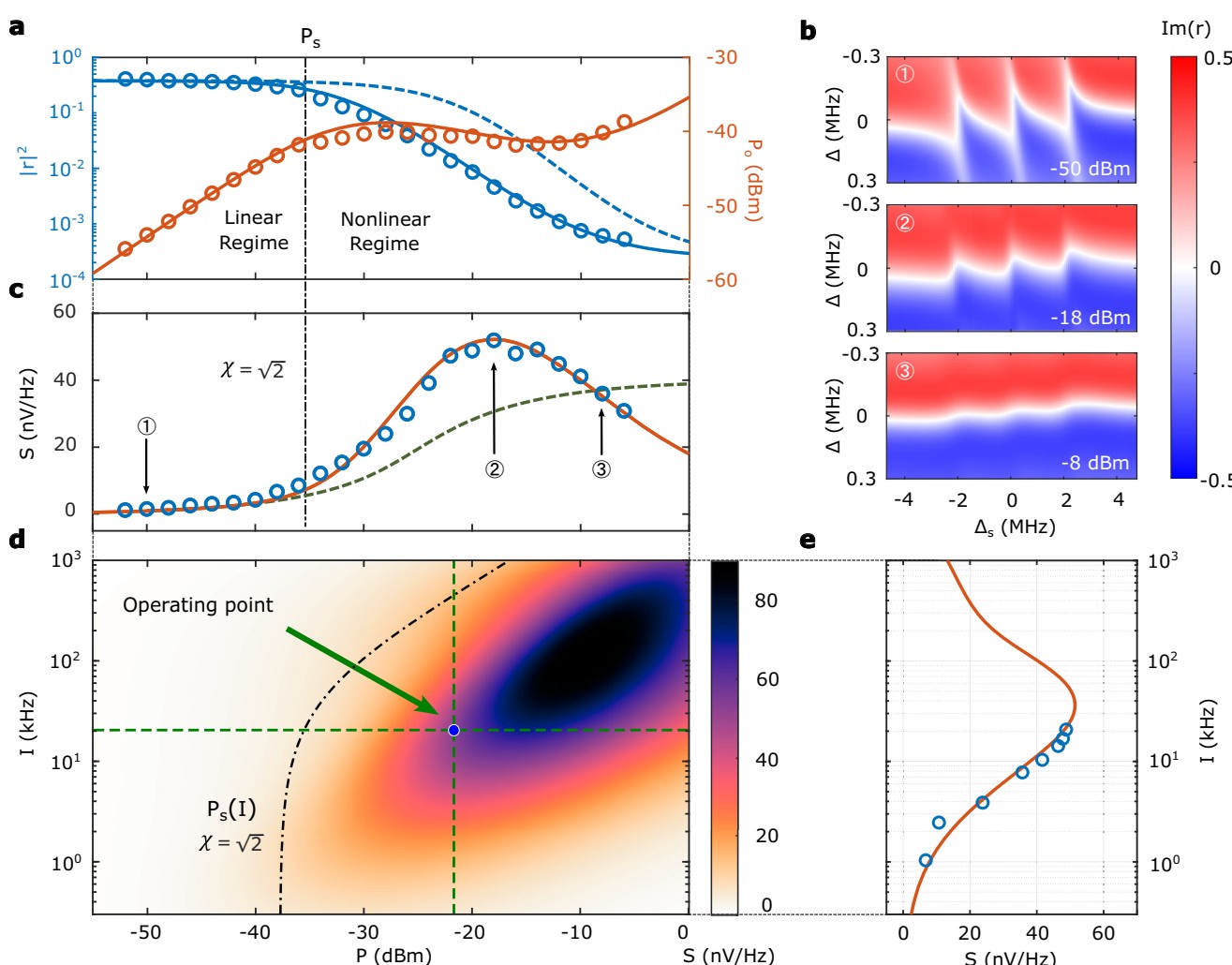

**Fig. 2 | cQED sensor nonlinear model. a** Power reflection coefficient $|r|^2$ (blue) and reflected power $P_O$ (red) with resonant tuning ($\Delta = \Delta_s = 0$). Circles are experimental data, solid lines are model results using parameters $\kappa_{c1} = 2\pi \times 125$ kHz, $\kappa_c = 2\pi \times 130$ kHz, $\Gamma = 2\pi \times 330$ kHz, $\gamma_0 = 2\pi \times 26$ kHz, $g = 2\pi \times 190$ kHz, $L = 0.53$. The optical excitation rate, I, is related to the effective longitudinal relaxation, $\gamma_p$ and the equilibrium polarization, $\bar{P}$, through the expressions given in Supplementary Note IID. The dashed line shows the model result assuming zero inhomogeneous broadening. The saturation power $P_s$ is plotted with the dash-dotted lines in **a**, **b**, **d**. **b** Experimental imaginary component of the cavity reflection coefficient as a function of probe ($\Delta$) and atom-cavity ($\Delta_s$) detunings for microwave input powers

$P = -50$ dBm, $-18$ dBm, and $-8$ dBm. **c** Device signal $S$ as a function of input microwave power at fixed optical polarization rate $\gamma_p = 2\pi \times 1.36$ kHz. Blue dots: experimental data. Red line: theoretical plot from nonlinear theory using the parameters given above. Green dotted line: the signal $S$ predicted by the spectroscopic signature model presented in[32]. The indicated points 1, 2, 3 correspond to the spectra shown in **b**. **d** Device signal $S$ as a function of input microwave power and optical polarization rate. The operating point that maximizes signal $S$ is indicated with the blue dot **e**, $S$ as a function of optical polarization rate at microwave power $P = -22$ dBm. The nonlinear model is the red curve and experimental measurements are blue points. The errorbar is smaller than the marker.

where $C_\alpha = 4g_{eff}^2/\kappa\Gamma_1$ is the effective cooperativity including the nonlinear saturation effect. Here $\Gamma_1 = \Gamma + \gamma\chi$ is the effective spin linewidth and $g_{eff} = g/\sqrt{\chi}$ is the effective coupling strength for collective coupling strength $g = \sqrt{\mathcal{P}Ng_s}$. At nonlinear microwave drive powers, these parameters are modified from their linear-regime values by the factor $\chi = \sqrt{1 + 8g_s^2|\alpha|^2/\gamma\gamma_p}$ that quantifies the depolarization of the spins due to the cavity field.

The cavity occupancy at which saturation begins has a natural interpretation: when the Rabi frequency of the resonant spins due to the cavity field $\sqrt{2}g_s|\alpha|$ becomes comparable to the geometric mean of transverse and longitudinal relaxation rates, the system behaves nonlinearly. For an initially polarized system, the input field at which saturation occurs is:

$$|\beta_s|^2 = \frac{Ng^2}{2\kappa_{c1}}\left[\frac{\gamma}{\sqrt{2}(\Gamma + \sqrt{2}\gamma)} + \frac{\kappa\gamma}{4g^2}\right]^2. \qquad (5)$$

The corresponding saturation power $P_s = \hbar\omega|\beta_s|^2/L^2$ is plotted on Fig. 2a, c, d. Following[32] we have introduced a fitting parameter, $L$, between the measured input power and the associated field incident on the cavity. We find a best fit with $L = 0.53$ using theoretical values for the bare spin coupling, NV absorption cross-section, and homogeneous dephasing rate given the specified NV density. This factor could arise due to microwave loss before the cavity coupling, or uncertainty in the theoretical parameters.

Around the discussed resonant tuning point, the reflection coefficient response is confined to the quadrature channel and is given by:

$$\frac{\partial \text{Im}[r]}{\partial\omega_s} = \frac{4\chi C_\alpha}{\Gamma_1}\frac{\kappa_{c1}}{\kappa}(1 + C_\alpha)^{-2}. \qquad (6)$$

This expression is the key result of our model; it allows us to interpret the observed optimal choice of optical polarization rate and probe power for our device, and to provide an outlook and recommendations for future device development. We plot the modeled signal response of our system as a function of drive power and optical polarization rate in Fig. 2d. In Fig. 2c, e, we plot cuts in microwave and laser power, respectively, through the optimal operating point, comparing model predictions (red curve) to our experimental data. Note that for weak optical illumination, the equilibrium polarization $\bar{\mathcal{P}}$ in Eq. (2) is suppressed while the longitudinal relaxation rate $\gamma_p$ approaches the intrinsic spin-lattice relaxation rate[40] (see Supplementary Note. II). We achieve a maximum signal response of 52 nV/Hz at $P = -18$ dBm and $\gamma_p = 2\pi \times 1.36$ kHz (point 2 in Fig. 2c), limited by the output power of our polarizing laser. The operating point considering power-dependent noise effects (discussed below) is at $P = -22$ dBm, indicated by the blue circle in Fig. 2d.

In contrast to previous efforts, this treatment of ensemble inhomogeneity and the optical polarization rate explains the value of the optimal microwave power and provides a better fit for our measured signal $S$ in the nonlinear regime. For comparison, we plot the magnetic sensitivity derived from the model presented in ref. 32 in Fig. 2c. This analysis calculates the expected polarization of the spin ensemble under the microwave drive and uses this to construct an effective linear Hamiltonian with a reduced, polarization-dependent coupling constant. This approach captures the quenching of the ensemble-resonator coupling at strong microwave drive, which is the focus of their study, but not the power-broadening of the ensemble due to the resonator field. We self-consistently determine the cavity field in the presence of the microwave drive and the spin ensemble, and then calculate the spectroscopic properties of the system from the cavity field. This accurately models the reduced signal in the strong probe regime and the optimal probe power shown in Fig. 2c.

## Noise analysis and spin refrigeration

Having characterized the sensor response $S$ in the section above, we now examine the noise environment and its impact on sensitivity. We measure the signal, isolated in the reflected quadrature, using a saturated mixer scheme (see Methods). Johnson-Nyquist noise typically sets the noise floor of microwave measurements. This voltage noise is characterized by $\mathcal{L}_{th} = k_B TR$, where $k_B$ is Boltzmann's constant, $R$ is the termination resistance, and $T \gg \omega/k_B$ is the device temperature. In addition to the Johnson-Nyquist noise, we also experience phase noise due to fluctuations of the probe field and added noise in the homodyne measurement chain (e.g. amplifier, digitizer; see Supplementary Note III).

In our system, the NV spins are polarized using continuous optical pumping. In this low entropy configuration, the spin ensemble serves as a cooling agent for the cavity mode by collectively interacting with microwave photons[28], an effect seen previously in e.g. Rydberg atoms[41]. This effect is plotted in the main panel of Fig. 3a. To establish a noise baseline, we detune the spin ensemble from the cavity by 5 MHz and measure the noise power spectral density in a 1 Hz band at 15 kHz offset in the signal channel for the bare cavity driven on resonance (blue). As we will show in Fig. 4, the noise floor at 15 kHz is dominated by the thermal noise but without the flicker noise components. At low microwave drive strengths, the microwave-power-independent thermal noise floor is characterized by $T = 407$ K, including a 0.8 dB amplifier noise figure. Above approximately $P = -15$ dBm, the thermal noise floor is eclipsed by the phase noise of the microwave drive (see Supplementary Note III).

Bringing the spin ensemble on resonance with the cavity (Fig. 3a, red), we find a 1.98 dB suppression of the thermal noise floor at low microwave probe power, which corresponds to an effective microwave temperature reduction of $\Delta T = 166$ K. In the inset of Fig. 3a, we plot the noise power spectral density at 15 kHz offset as the microwave drive is tuned through resonance with the cavity and spin ensemble. Despite the presence of a strong drive (150 dB greater than the noise), the cooling effect persists in the spectral region of interest. The cooling and resulting sensitivity are shown as functions of probe power in Fig. 3b.

We find good agreement between measurements and a nonlinear cooling model (see Supplementary Note III), where the bare cavity parameters are adjusted to effective parameters using the nonlinear factor $\chi$. This spin refrigeration effect begins to be suppressed at the same saturation onset power $P_s = -35$ dBm described in Eq. (5) as the effective coupling strength is diminished, with a reduction by 1/e at $-23$ dBm. Oscillator phase noise also contributes to the overall noise performance in this region, increasing both the cooled (red) and uncooled (blue) noise floors. Around the sensitivity-maximizing operating point $P = -20$ dBm (note that this is different from the signal optimal point $-18$ dBm) we observe a noise thermal power reduction of 0.44 dB, demonstrating that spin refrigeration offers sensitivity enhancement for an actively-driven system.

## Broadband Magnetometry

Next we examine the performance of the device in the frequency domain. We operate at a fixed power of $-22$ dBm which produces near-optimal sensitivity in the thermal-limited regime as described above, while also reducing the contributions of power-dependent flicker and phase noises (see Supplemental Note III). In Fig. 4, we plot the magnetic sensitivity in blue across a spectral range of 10 Hz–100 kHz. The mean sensitivity of a 1 Hz band between 10 and 11 kHz offset is $\eta = 576$ fT/$\sqrt{\text{Hz}}$ with a standard deviation of 6 fT/$\sqrt{\text{Hz}}$, and does not roll off at high frequencies within the measurement bandwidth of 100 kHz. This is in contrast to sensors using e.g. dynamical decoupling techniques[2] or with a limited lock-in bandwidth[3] that have significantly reduced spectral response. The inset of Fig. 4 shows the sensitivity near 20 kHz with an amplitude spectral density derived from the

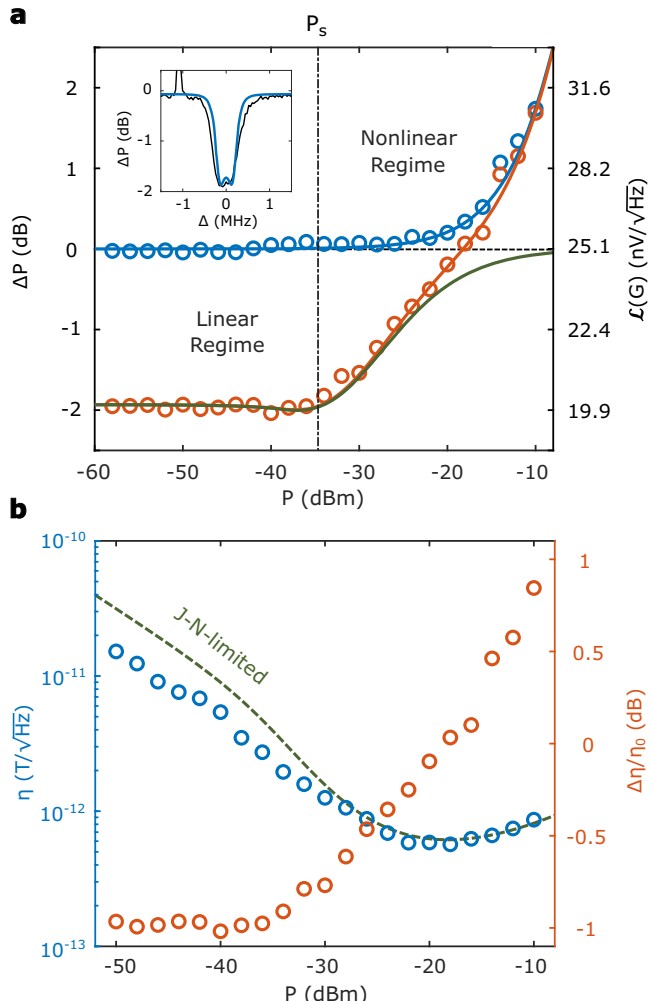

**a**

**b**

**Fig. 3 | Spin refrigeration. a** Noise reduction with different microwave driving powers *P*. Blue (red) dots show the noise in 1 Hz bandwidth at 15 kHz offset in two regimes of spin-cavity detuning $\Delta_s \neq 0$ ($\Delta_s = 0$). The left axis shows the relative change in power spectral density compared to a 50 Ω reference measurement, and the right axis shows the voltage amplitude spectral density. All measurements were taken with a power gain of 36.5 dB and the values are doubled-sided. The blue curve shows the noise model considering thermal and phase noises. The red curve shows the noise model with linear steady-state cooling (See Methods). The green curve is the contribution of mode cooling alone, without thermal or phase noises. Inset: frequency-resolved noise reduction with $P = -58$ dBm and $\Delta_s = 0$. **b** Sensitivity $\eta$ at 15 kHz offset with different microwave power (blue dots). The blue curve shows the sensitivity corresponding to the room temperature limit $\eta_0$. The red dots show the sensitivity difference $\Delta\eta/\eta = (\eta - \eta_0)/\eta_0$. We observe sub-thermal sensitivity with $P < -18$ dBm. The errorbar is smaller than the marker.

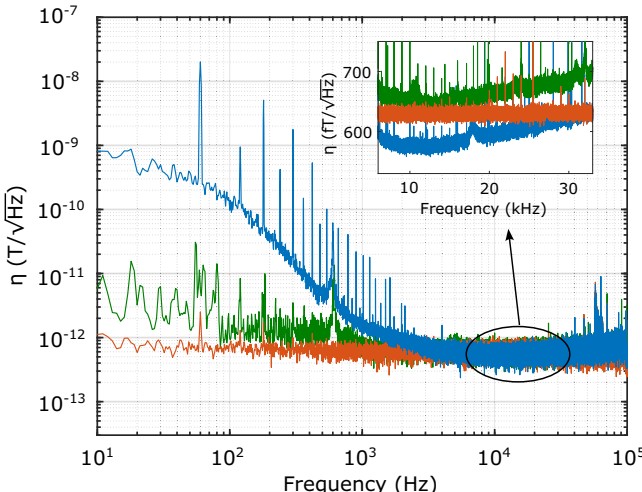

**Fig. 4 | Broadband magnetometry.** Sensitivity of the cQED sensor as a function of magnetic field frequency derived from the voltage noise floor of the device. Red: The magnetic-field-equivalent noise floor of the microwave circuitry, including low noise amplifier, circulator, and mixer, measured by replacing the cQED sensor with a 50 Ω terminator. Green: The field-equivalent noise floor of the cQED sensor with NV spins detuned ($\Delta_s = 5$ MHz). Blue: Magnetic field sensitivity at the optimized operating point. Inset: the magnetic field sensitivity is around 20 kHz showing the mode cooling effect. We achieve a sub-room-temperature Johnson-Nyquist limit sensitivity of $\eta = 576 \pm 6$ fT/$\sqrt{\text{Hz}}$.

spectrum in the presence of a test magnetic field of known amplitude. This and further characterization of broadband sensor linearity and amplitude dynamic range is presented in the Supplementary Note IV.

## Sensitivity outlook

Finally, we discuss the implications of these results for the design of future devices. The diamond is characterized by its shape, volume $V_d$, and NV density $\rho$. The NV ensemble inhomogeneous linewidth and intrinsic homogeneous linewidth are linearly related to the NV density[42], i.e. $\Gamma = a\rho$ and $\gamma_0 = b\rho$ with coefficients $a = 2\pi \times 82.5$ kHz/ppm and $b = 2\pi \times 7.5$ kHz/ppm consistent with our device. Here $\gamma_0$ is the intrinsic homogeneous linewidth in the absence of the optical polarization cycle (see Supplementary Note IID). The resonant cavity mode is similarly characterized by its shape, volume $V$, and quality factor $Q = \omega_c/\kappa_c$. The cavity mode volume implicitly sets the single-spin coupling strength, $g_s$, which, together with the ensemble size, $N = \rho V_d$, and equilibrium optical polarization, $\bar{\mathcal{P}}$, determines the collective coupling strength, $g$.

We first consider the optimal diamond for our present microwave cavity mode, $V = 1.7$ cm³ and $Q = 2.2 \times 10^4$. Increasing the diamond volume at fixed NV density improves device sensitivity up to unity filling factor $V_d/V = 1$, as shown in Fig. 5a. In the low-$\rho$ regime ($\rho < 0.02$ ppm), laser-induced dephasing dominates both the inhomogeneous and homogeneous NV linewidth; therefore, adding more density to the system increases the collective coupling without the cost of broadening the NV spins. In the medium-$\rho$ regime (0.02 ppm < $\rho$ < 3 ppm), the density-induced inhomogeneous broadening eclipses the laser-induced dephasing, and the sensitivity is almost independent of density $\rho$. In the high-$\rho$ regime ($\rho > 3$ ppm), the spin dipole-dipole homogeneous dephasing starts dominating the laser-induced dephasing, causing a sensitivity drop in this domain.

Larger diamond volumes yield better sensitivity but can be more challenging to optically polarize as the incident laser is attenuated within the diamond. The optical polarization constraint $\rho_{max} V_d = 0.49$ cm³ ppm (See Supplementary Note V) is satisfied to the left of the black dotted line in Fig. 5a, assuming a diamond aspect ratio of 2.2 as in our

average of 3600 independently measured power spectral densities. In this region, device sensitivity improves on the 620 fT/$\sqrt{\text{Hz}}$ noise-equivalent sensitivity implied by the standard 50 Ω termination (red)−beating the thermal Johnson-Nyquist limit. This sensitivity, along with the associated inverse readout fidelity of $\sigma_e \sim 360$, both set the standard for the highest-performing continuous operation NV sensor.

At frequencies below 1 kHz, ambient magnetic fields dominate the low-frequency noise and set the noise floor. With the spin ensemble off-resonance, the sensor magnetic response is negligible, and low-frequency environmental magnetic fluctuations are suppressed in the bare cavity noise-equivalent sensitivity (green). From this spectrum, we infer a sensitivity of $\eta \approx 2$ pT/$\sqrt{\text{Hz}}$ at around 15 Hz, neglecting possible low-frequency magnetic fluctuations intrinsic to the diamond. To validate our noise-spectrum-inferred sensitivity, we record a noise

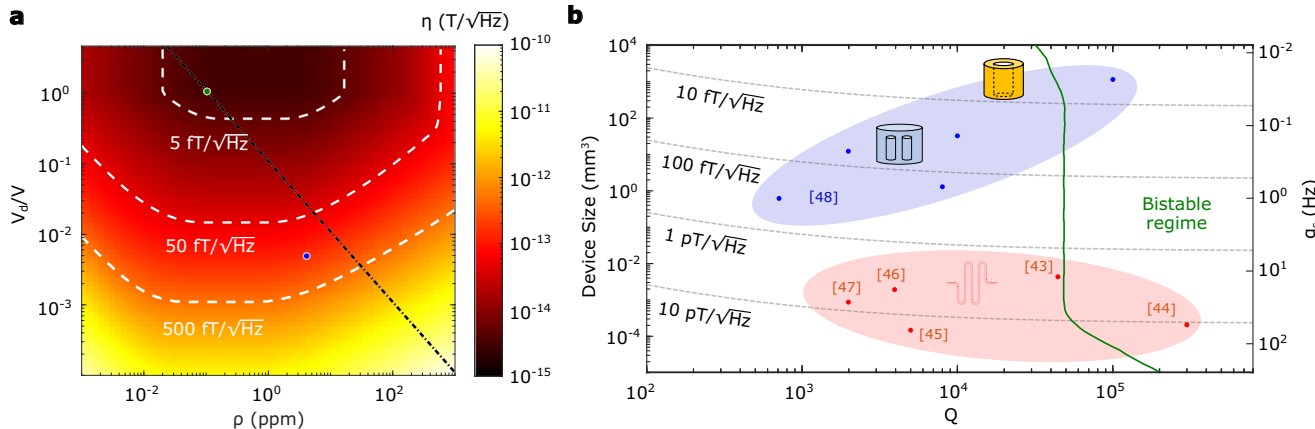

**Fig. 5 | Cavity and spin optimization predicted by the nonlinear model.**
**a** Magnetic field sensitivity with different diamond volumes $V_d$ and NV density $\rho$. The diamond volume is normalized by the cavity mode volume ($V = 1.7$ cm³). Three dotted lines show the contour of the sensitivity $\eta = 5$ fT/$\sqrt{Hz}$, 50 fT/$\sqrt{Hz}$, and 500 fT/$\sqrt{Hz}$. The blue dot shows the current device and the green dot shows the best sensitivity achievable from an optimized diamond using the cavity shown in

this work. The black dotted line shows the optical polarization limit. **b** Magnetic field sensitivity with different unloaded quality factors $Q$ and single coupling strength $g_s$. The red dots show cavity design for stripline cavities or superconducting circuits[43–47]. Blue dots various three-dimensional cavity designs from[48]. The current device is shown as a yellow cylinder.

current experiment. Larger aspect ratios could relax this constraint and potentially achieve better sensitivity but at the cost of higher laser power. In this analysis, we limit the laser power to a maximum of 20 W. An optimal diamond (green dot) would fill the cavity mode and have the largest possible density of NV centers while still allowing polarization of the full ensemble. We predict that the resulting device should have a sensitivity of approximately 3 fT/$\sqrt{Hz}$.

We next discuss alternative cavity designs with different $Q$ ($\kappa_c$) and $V$ ($g_s$). We assume that all cavities are filled with the diamond at the unit filling factor, $V_d = V$, and optimize the sensitivity of the device at each $Q$ and $V$ under the constraints described above. The results are shown in Fig. 5b, with realistic cavity examples indicated as dots[43–48]. Two trends are clear in the results shown in Fig. 5b. The optimal sensitivity $\eta$ is almost independent of $Q$. This scaling is in contrast with previous predictions that sensitivity would be proportional to device cooperativity (linear in $Q/V$). In the saturated regime, the effective cooperativity $C_\alpha$ is independent of $Q$ as the reduced cavity linewidth is compensated by increased spin linewidth as well as reduced effective coupling. An optimized device could reach $< 10$ fT/$\sqrt{Hz}$ in the high-$Q$, large mode limit (upper right region of Fig. 5b). Secondly, smaller cavities can achieve efficient readout, but have lower overall sensitivities as their spin ensemble sizes are reduced with the assumption of unity filling factor.

## Discussion

The demonstrated magnetic sensitivity of $576 \pm 6$ fT/$\sqrt{Hz}$ sets the standard for the highest-performing continuous-operation NV sensor[2,25,27]. The adaptation of additional sensitivity-improving techniques into the cQED platform, such as pulsed operation e.g. echo sequences, P1 bath driving, and double quantum transitions[49], or the use of flux concentrators[13], could further improve sensitivity.

Spin refrigeration enables high performance beyond the thermal noise that was previously thought as the fundamental limit for ambient operation, and offers a path towards quantum-limited devices at room temperature. Our theoretical framework on the inhomogeneous nonlinear theory, in agreement with the experiment, also allows further optimization and understanding of the operation of cQED sensors in the highly cooperative regime. Future research will explore several avenues enabled by this proof of concept: the extension of our inhomogeneous, nonlinear theory to sensors in the bistable regime, optimization of spin refrigeration toward the quantum limit, and the exploration of other sensing modalities, such as inertial sensing and timekeeping applications.

## Methods
### Experimental setup
We employ a saturated mixer scheme to measure the phase change induced by the NV-cavity system. Probe microwaves from a signal generator are divided into a reference arm and a signal arm using a Wilkinson microwave power divider. The reference arm is directly connected to the LO port of a mixer (HX3400), with a voltage-controlled phase shifter to tune the relative phase. The signal arm is directed to a circulator with tunable attenuation for power control on the cavity input, and the circulator's microwave output is coupled into the dielectric resonator using a probe loop. The reflected signal from the cavity returns to the circulator and is connected to a low-noise amplifier and subsequently the RF port of the mixer for a homodyne measurement. This setup effectively separates the reflected microwave signal from the incident signal, enabling the measurement of the quadrature part of the reflection coefficient by an appropriate setting of the LO phase shifter. The quadrature signal is subsequently digitized using a sampling rate of 200 kS/s. Power spectral density in Figs. 3 and 4 are calculated with Welch's power spectral density estimate. For power measurements (Fig. 1c), an IQ mixer is used to measure both field quadratures, and the total power is then computed.

The diamonds (3 mm × 3 mm × 0.9 mm, 4 ppm NV ensemble, sourced from Element 6) are set at the TE01$\delta$ mode maximum point in the center of the dielectric resonator (Skyworks, $\varepsilon_r \sim 31$, center frequency: 2.877 GHz). A wafer of 4H-SiC is used for heat transfer and supporting the diamond, while two pieces of low-loss-tangent polytetrafluoroethylene (PTFE) are used to fix and align the dielectric resonator. An aluminum shield is employed to isolate the system from external signals in the lab, such as WiFi and 3G signals (1.9 GHz), and to reduce radiative losses. An 8 W 532 nm pump laser is utilized to optically polarize the spin ensemble. External magnetic bias is provided by 3-axis magnetic coils. Magnetic field along [100] direction has an amplitude of $\sim 4.3$ G to move the NV resonant frequency to be on resonance with the cavity frequency.

### Test field
We performed a test magnetic field measurement using a coil with turn number $N = 400$, radius $r = 7$ cm, and distance between the coil and sensor $d = 22$ cm. We use a DC power supply to generate a voltage $U = 1$ V, resulting in a current output of $I = 43$ mA. The magnetic field generated by this test coil at the sensor position is $B_{cal} = 2\mu_0 \pi r^2 NI/4\pi(d^2+r^2)^{3/2} = 4.3$ μT, which is similar with the

measurement from the measurement from a commercial gaussmeter $B_{Gauss} = 3.9$ µT. Our sensor shows a response of $B_{cQED} = 4.0$ µT with a 2.5% (7%) error compared with the reference magnetometer (calculated result). The difference can be attributed to the estimation of the size parameters and the twisted angle of the coil with the diamond surface. An AC test magnetic field is also shown in the Supplementary Note IVc.

## Sensitivity prediction

We applied the first-order approximation (See Supplementary Note II) for the sensitivity optimization for Fig. 5. We then collected the optimal parameters and determined the number of solutions. Following this, we identified and marked the bistability regime in Fig. 5b, which is characterized by its multiple solutions. The parameters for optimization and the constraints are listed and plotted in Supplementary Note IV.

## Data availability

The data presented in Figs. 1–4 are available at the following Git repository: https://github.com/hanfengw/cQEDsensor

## Code availability

The code that supports the findings of this study is available from the corresponding author upon request.

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

## Acknowledgements

The authors would like to thank Donald Fahey, Reginald Wilcox, David Phillips, Danielle Braje, Andrew J. Kerman, and Avetik Harutyunyan for helpful discussions. H.W. acknowledges support from Analog Devices, Inc. and Honda Research Institute USA, Inc. D.R.E. acknowledges funding from the MITRE Corporation and the U.S. NSF Center for Ultracold Atoms. This material is based upon work supported by the Dept. of the Army and the Under Secretary of Defense for Research and Engineering under Air Force Contract No. FA8702-15-D-0001. Any opinions, findings, conclusions or recommendations expressed in this material are those of the author(s) and do not necessarily reflect the views of the Dept. of the Army and the Under Secretary of Defense for Research and Engineering.

## Author contributions

H.W. and M.E.T. created the setup and conducted the experiments. D.R.E. proposed the spin refrigeration for the cQED sensor. H.W., K.L.T., K.J., D.R.E and M.E.T. developed the nonlinear model. H.W., K.L.T, and M.E.T. prepared the manuscript. X.Z. and M.J. assisted in electrical measurement design. All authors discussed the results and revised the manuscript. M.E.T. and D.R.E. supervised the project.

## Competing interests

The authors declare no competing interests.
