## [Transparent Peer Review file · Nature Communications]

A spin-refrigerated cavity quantum electrodynamic sensor

Corresponding Author: Dr Matthew Trusheim

Version 0:

Reviewer comments:

Reviewer #1

(Remarks to the Author)

In the current manuscript, the authors reported the optimization of the cavity-enhanced microwave readout of the NV ensemble-based magnetometer at room temperature as demonstrated in their previous study [Eisenach, E. R., et al. Nat. Commun., 12(1), 1357 (2021)] by study experimentally and theoretically the influence of the microwave probe and the optical spin polarization. Under the identified optimal condition, the minimal sensitivity is reduced from 3 pT/√Hz to 0.58 pT/√Hz, i.e. an enhancement of five times. During the optimization of their device, the authors identify two interesting phenomena, i.e. microwave power-induced transition of the split reflection spectrum to the single peak reflection spectrum, and the microwave mode cooling. Although these phenomena have been well studied before both experimentally and theoretically, the authors demonstrated here for the first time that they can be used to improve the sensitivity in the CQED-based quantum sensing at room temperature. In the supplementary material, the authors demonstrated even the sensing of an AC magnetic field with an amplitude as low as 5 nT.

The current work presents indeed some new results which deserves the publication. However, by reading the manuscript, I was confused by the presentation of the contents, and have also encountered numerous typos or mistakes, which might degrade the readability of the manuscript. Thus, I would like to suggest the authors to revise systematically their manuscript before publishing it.

Comment 1: It seems to me that the title might need to be improved to reflect the sensitivity optimization via the two phenomena mentioned above.

Comment 2: Although the theoretical modelling reproduces very well the experimental results, it is quite difficult for readers to perceive the transition from the linear regime to the nonlinear regime. Can the authors come up with some schemes to illustrate the physics behind?

Comment 3: It seems that the influence of the microwave cooling is only visible by averaging over tens of measurements. Can the authors provide an explanation for this?

Comment 4: The authors might need to provide some schematics to illustrate the different kinds of measurement setups, which will be helpful for the explanation of the noise analysis.

Comment 5: The manuscript might need to be reconstructed. Since the paper aims to optimize the sensitivity through the microwave probe and the optical polarization, it might be necessary to give the expression for the sensitivity and the signal in the very beginning, and indicates then that the signal should be optimized through the nonlinear model of the reflection coefficient.

Comment 6: It might be necessary to provide the line-cuts for $\Delta_s = 0$ in Figure 2b to illustrate the shape of the imaginary part of the reflection coefficient.

Comment 7: The thermally-polarized exceptional point model should be explained in more details. It is not clear what is the difference between this model and the model used in the current study.

Comment 8: The optical polarization is indicated sometimes as γ_0 , like Figure 2d and e, but is denoted sometimes as γ_p . The authors should make sure the consistence of the symbols.

Comment 9: In section III A, it is said that the sensitivity-maximizing operating point is -22 dBm. However, the discussion in the previous section seems to indicate a power -18 dBm. The authors need to verify the value.

Comment 11: If we assume $g_s = 0$ in the last expression of Eq. (2), we will get an analytical solution to the steady-state population p_j , which decays exponentially to zero. It means that independent of the laser power, all the spins are polarized to the ground state, which does not agree with what is expected. This problem occurs since the authors do not account for the spin-lattice relaxation in a proper way. In some part, the author argued that the intrinsic spin relaxation is ignored because it is smaller than the optical spin polarization rate and the microwave-induced depolarization rate. However, the spin-lattice relaxation amounts to about 200 Hz at room temperature, which is not too small than the minimal optical polarization rate 1kHz. I suggest the authors to check the models in the article [Zhang, Y., et al., Phys. Rev. Lett., 128(25), 253601 (2022)], and model the spin-lattice relaxation more accurately.

Comment 12: Below equation (S2c), it is said that p_j is the population on the state $|+1\rangle$. However, later on, it is said that it is the population difference. The authors should check through the manuscript to ensure the consistent interpretation of the same term.

Comment 13: Below the equation (S6), it is said that the population is perfectly initialized to $|0\rangle$ state for $|\alpha| = 0$, under the assumption of large NV polarization rate γ_0 . However, it seems to me that γ_0 is the spin relaxation rate, while γ_p is the optically induced polarization rate.

Comment 14: In the equation S10, lots of new terms are introduced, and some are not explained immediately. What is the function of h ? What is Γ ? In addition, there is a large gap between Eq. (S9) and Eq. (S10). The authors need to explain better.

Comment 15: In the supplementary section H, Eq. (48) is described as the equations for the atomic coherence and the atomic inversion. The authors should change "atomic" by "spin". In addition, p_j in Eq. (48) describes the population inversion, while it is said as the population on the $|+1\rangle$ spin states before. Such typos or mistakes should be corrected.

Comment 16: How is the polarization defined in Eq. (55)? For large pumping, it is about $(1-\eta)/(1+2\eta) \approx 0.32$ with $\eta \approx 0.4$. Such a polarization seems to be too small for me. The polarization about 80% is normally reported in literature.

Comment 17: In the last part of the supplementary material, the different kinds of noise are discussed. It might be helpful if the authors can provide a schematic for the homodyne measurement setup.

Comment 18: It seems that the caption for the Supplementary Figure 6 and 7 are mistakenly exchanged. The authors should correct this mistake.

Reviewer #2

(Remarks to the Author)

I have sectioned off my comments to focus on the aspects highlighted in the reviewers guide for Nature publishing group articles.

Key results

The Authors demonstrate a room temperature magnetic field sensor that utilises an ensemble of NV centres strongly-coupled to a microwave cavity. The Authors spin measurement is based on a readout of microwave photons rather than probing the optical transitions in a more typical ODMR setting. These microwaves are used to directly probe the ground state spin sublevels of the NV ensemble. By doing this they are able to remove the complexity of designing an optical cavity to enhance their measurement sensitivity. Instead, they can use microwave cavities where it is significantly easier engineer a cavity that can reach the strong coupling regime. The notable claim seems to be a record sensitivity for magnetometry using an NV centre, which is an interesting result and one that others will look to use as a reference point as a benchmark for comparison in the future. The results seem to be built on several innovations namely the strongly coupled microwave cavity, the "spin cooling", and control of the microwave/optical-polarisation power. I have some questions surrounding which of these are new to this manuscript (see clarity and context)

Clarity and context

The big conclusion seems to be that "The demonstrated magnetic sensitivity of 580 fT/√Hz sets the standard for the highest-performing continuous operation NV sensor". To the best of my knowledge this seems to be the case. The nearest being the 900fT/Hz found in reference [13] Ilya Fescenko, et-al, Phys. Rev. Research 2, 023394 (2020), which is not specifically called out during the concluding remarks of the paper. Further, I think the quoted statement is a little ill defined. For example, the use of "continuous operation" looks to me like a qualifier. Is this setup unequivocally the most sensitive NV-diamond magnetometer? Is there some pulsed operation that is more sensitive? Also, there is no context to this statement, the authors should more explicitly state how their results compare to those of others. I suggest providing a direct quantitative comparison to allow the reader to easily interpret the advances achieved.

The work presented in this manuscript seems to build on previous publications specifically: reference [24] E.R.Eisenach, et

al, Nature Communications 12,1357 (2021), and reference [25] D.P. Fahey, K. et al, Phys. Rev. Appl. 20, 014033 (2023). Reference [25] is clearly from the same group of academics, and I think the same for [24] although this is less clear. I think the authors should be more explicit and take ownership of their previous work just to make it clear what they themselves have done previously. This is problematic particularly in the abstract where they state “Here we move towards this quantum limit... by introducing (i) a cavity quantum ... (ii) a comprehensive non linear model... (iii) spin refrigeration”. (i) was introduced in ref [24] and (iii) in [25]. I think the authors should put more focus on the specific innovations present in this manuscript whilst acknowledging how it builds on their previous efforts. My understanding is the main innovation is due to the control and modelling of the MW power/optical spin-polarisation rate by moving into the non-linear regime? I think this could be communicated more effectively with the reader.

It needs to be made more explicit what new innovations specific in this manuscript have led to the increased sensitivity. I also think the authors should place the results in wider context by comparing them with other magnetic sensors including SQUID and atomic based. I appreciate comparisons can be tricky and each implementation will have unique advantages and disadvantages. However, as it stands the non-specialist reader will not be able to interpret the significance of the implementation in this manuscript.

More specific clarifications:

Why does changing the density of NV centres not impact the sensitivity whereas the volume of the Diamond does? This doesn't make intuitive sense to me and I would appreciate some clarification.

Section III, B related to broadband operation does not quantify what counts as broadband. This could be remedied via comparison with other results in both diamond colour centres and indeed other materials/systems. Once again someone who is not familiar with sensing applications will have no context for the claim of broadband operation.

Validity

The analysis of the results all looks correct as far as I can see. The headline claim of a sensitivity of $580 \text{ fT}/\sqrt{\text{Hz}}$ is backed up by the data. Although it is unclear why the result is stated to this level of precision, the data in Fig.4 looks a little noisy. I would say it is $\sim 600 \text{ fT}/\sqrt{\text{Hz}}$ based on the data in Fig.4. Perhaps some error on this sensitivity could be included?

Significance

I am left a little unclear on the significance of what has been achieved in this manuscript due to lack of clarity on the main innovations that are unique to this manuscript, rather than those in previous publications (refs [24], [25], see also clarity and context)

I do think the headline result for the sensitivity quoted in this manuscript will be of interest to researchers in the field of sensing with defect centres in diamond. It will undoubtedly function as a benchmark that will be used for others working in the field. I think there is the potential for interest from non-specialist readers, however the way the work is currently presented in this manuscript will dampen the appeal (see clarity and context, suggested improvements).

Data and methodology

The reflection spectra in Fig 1.c looks very similar to the one in ref [25] apart from the fact the y-axis has been inverted. Since the authors have not properly acknowledged their previous work it is not possible to know if this data is new to this manuscript, i.e., is it the same sample, same experimental conditions, if it is different how is it different? Incidentally similar data is presented in Fig.2 b (for different MW powers) where the y-axis is inverted back to align with the convention in their previous work. A consistent presentation format for the data would be better.

Analytical approach

This all looks sound the authors are using fairly standard CQED formalism and are using techniques established in previous publications.

Suggested improvements

I do not find the energy level diagram in figure 1b particularly illuminating. What is happening between the left- and right-hand side of this diagram? is it illustrating the effect of hyperfine coupling with the ^{14}N ? I think the experiment focusses on transitions between the 0 and +1 ground state manifolds (ignoring the -1), but I can't find where this is explicitly stated. Also, the little balls with arrows that are intended to represent spins (I presume) all point in the same direction is this correct/intentional? If so it is maybe not a good way to differentiate between the transitions. The curved grey lines I think represent the cavity, but these are drawn around a specific transition, implying the others do not couple, which I don't think it the authors intention. The caption for this also needs more information to help clarify this sub-figure.

In general I find the manuscript a little uninspiring It reads more like thesis chapters. It is quite disjointed and does not flow nicely. I was frequently bogged down in details and found it challenging to follow the main thread of the paper. This makes it even more challenging to unpick the innovations and significance of the result. As I have said the authors should highlight the main innovation specific to this paper that has enabled their result and build the manuscript around this. The reader

should be provided sufficient detail to understand the result/innovation. I encourage the authors to consider what information is essential for the reader to understand and interpret the results. If it is not absolutely necessary then it can be moved into the supplement. Also if there are details in previous manuscripts it will be sufficient to summarise the main findings to the reader and redirect them to the relevant publications for further details.

Further issues can be solved by spending more time placing the results in context, to better engage the reader. I think the main innovation here is a detailed study of how controlling the microwave/optical power can increase the sensitivity. The paper jumps immediately into the theoretical CQED background which I find quite jarring. A lot of the content in section II is covered in previous references and much of it looks like standard CQED theory. The non-linear part looks like the main innovation. I wonder if there is a better way to discuss this that focusses more on the results and shows how this is backed up by a detailed theoretical model. In this case much of the CQED theory can be moved to the supplement.

References

The references section is fairly comprehensive. My only suggestions would be to include extra in response to areas I have highlighted where the manuscript lacks context.

My expertise

Quantum photonics with quantum emitters, spin-photon interfaces, single photon sources, CQED and open quantum systems and quantum technology in general. Familiar with magnetometry using NV centres in diamond, but not a specialist in this area.

Summary

Overall I am somewhat torn for my recommendation for this paper. On the one hand the sensitivity achieved will act as a benchmark. However, this will likely be surpassed by others, including the authors at some point. However, I do acknowledge it is important to have these results published as a reference point. I will leave this as a decision for the editors.

The main problem is the way the manuscript is written is uninspiring and not typical of what I would read in nature communications. I think some significant re-writing will be necessary to improve the accessibility of the manuscript such that it can have the greatest possible impact.

Reviewer #3

(Remarks to the Author)

General

- the authors use an RF resonator and MW detection setup to read the spin states of NV- centers in diamond in the RF domain (rather than in the optical domain as is commonly done). using this approach, they claim superior readout of the spin states, reaching high sensitivity to changes in the magnetic field down to fT/\sqrt{Hz} .
- the authors use the power-dependent behavior in the strong-coupling regime to balance the readout photon numbers with the sharpness of spectral features to achieve optimal readout efficiency.
- the scientific methods seem adequate.
- the authors use two interesting methods: the first being the mode cooling shown by same authors in previous paper (Fahey, D. P., Jacobs, K., Turner, M. J., Choi, H., Hoffman, J. E., Englund, D., & Trusheim, M. E. (2023). Steady-State Microwave Mode Cooling with a Diamond N-V Ensemble. PHYSICAL REVIEW APPLIED, 20, 14033. <https://doi.org/10.1103/PhysRevApplied.20.014033>)

and second is the optimization of the MW and laser power to achieve optimal operation of the 'device'. however, this seem to be a small improvement compared with the results described in their previous work, and stands on well-known base, notably the power-broadening of NV centers, and nonlinearity of such experimental schemes (such as cited in the main text) therefore I am not convinced it is of great importance and significance.

- as a 'device' or 'sensor' author statements seems somewhat 'overselling':

the authors use a 8W green laser to pump the system, and still need hours-long measurements to achieve the claimed sensitivity, and need specialized electronics,

so while the sensitivity is impressive, it is not quite a 'device' at this point.

- magnetic field sensitivity: works in optically excited NV centers easily reach sensitivities around 1nT, but this is done with optical powers circa 1mW and on ensembles of <10 NV centers; and potentially with sub-micron spatial resolution.

In the system proposed here authors show averaging on a huge ensemble of NV centers and high laser powers, with 3 orders of magnitude increase in accuracy, but loss of spatial resolution.

therefore, I ask to see a figure-of-merit combining parameters such as number of photons in the system and their lifetime(optical and MW), number of NV centers, measurement time, etc, to know if there is really an improvement or is it merely taking advantage of averaging on a huge ensemble.

- highly specific paper, perhaps more fitting to an applied or basic physics journal.

Specific comments.

- the claim of 'first strong coupling in room temp' (lines 165-166) is false: it was achieved previously by the same authors in

the paper mentioned above.

- the text should explicitly declare the type (model) and exact design of the cavity, and include a measurements of the cavity's RF response (can be in supplementary).
- a sketch of the experimental setup (including MW circuitry and instrumentation) is needed in the supplementary information.
- system description: uniform coupling between each NV and the cavity mode is a valid assumption only if the mode peak area \gg NV ensemble size. cavity details not given so cannot judge the validity of this assumption.
- figure 1C and related text: what is Δ_s ? is it related to the Δ_j in the main text (line 105) ?
- more on figure 1C: the choice of parameters is not trivial for non-expert readers: it must be stated that the tuning of the NV resonances relative to cavity resonance is done by an external magnetic field which is applied like is done in ODMR experiments, I did not find this in the text, and it should be drawn in figure 1.
- figures 1C and 2A disagree on the amount of reflection $|r|^2$ for zero detuning: figure 1C suggests near zero reflection while figure 2A suggests reflection of $\sim 20\%$ (0.2). which is it ? does the cavity really offer near 0 reflection on it's resonant frequency ? what is the true reflection when on NV resonance ? is figure 1C normalized ?
- figures 1C and 2d use opposite color coding which I find confusing.
- line 49-410 : claims of applications in time-keeping, etc seem pre-mature, this experiment is very far from state-of-the-art in e.g. atomic clocks in both accuracy and integrability.

Version 1:

Reviewer comments:

Reviewer #1

(Remarks to the Author)

In the current work, the authors had not only made a great effort to optimize the microwave-based readout scheme proposed in 2021 [Nat. Commun 12:1357 (2021)], but also provided a comprehensive study on the response of the NV center spins-microwave cavity system to the microwave power and laser illumination. In particular, the authors have demonstrated for the first time that the microwave mode cooling, as revealed by several studies in recent years, can be utilized to increase sensitivity of magnetometer. Although not fully exposed in the main text, the theoretical treatment of the inhomogeneous broadening and the strong driving is not trivial at all. What more important is that the authors have made a great effort to achieve the agreement between the experiment and the theory. Based on these reasons, I recommend the publication of the paper.

In comparison to the last version of the manuscript, the authors have made significant efforts to address my and other reviewers' comments and to improve the readability. I am satisfied with all their responses. By reading through the revised manuscript and supporting information, I have also spotted some minor problems, and indicate them below.

Comments for the main text:

Comment 1: It seems to me that the authors have emphasized the spin-refrigeration more than the non-linear model in the title and introduction. However, in the text, the authors have explained first the non-linear model, and then addressed the spin-refrigeration. It seems to me that the spin-refrigeration is also part of the results from the non-linear model. Thus, I suggest the authors to revise some parts of the title and introduction to reflect the relation between the non-linear model and the spin-refrigeration effect.

Comment 2: It seems that there are too many symbols in the caption of Fig. 1, and they might be overwhelming to the readers. There are some symbols not explained, for example $\alpha, \omega_c, \omega_d, \omega_s, \beta$. Some of the symbols such as Γ, γ do not appear in the figure. The caption might not be the right place to introduce them. The energy diagram shown in Fig. 1(b) is too simplified, and the indication of the level transitions is not correct. It might also be necessary to explain what is the strong coupling features.

Comment 3: In the main text, it might be necessary to specify also the magnitude of the applied magnetic field, and the frequency of the cavity and the spin transitions. This information is necessary if someone wants to reproduce the results in the paper.

Comment 4. The author might need to explain where the factor $\sqrt{3}$ in Eq. (1) comes from.

Comment 5: In Fig. 2(a,c,d), the symbol $\chi=\sqrt{2}$ is indicated besides the dash-dotted lines. However, in the caption of Fig. 2, there is no explanation of this expression and the dash-dotted lines.

Comment 6: By looking carefully Fig. 3(b), one can see that the optimal sensitivity is achieved by a microwave power around -18 dBm rather than -22 dBm. The optimal microwave power -18 dBm is consistent with the power to achieve the largest signal, see Fig. 2c. By examining carefully Fig. 3a, it seems that the microwave noise starts increasing due to heating for the microwave power larger than around -22 dBm. However, the spin refrigeration compensates partly the heating, and improves the sensitivity. Thus, the power -22dBm can only make sense as the power to avoid the increased noise due to the microwave heating.

By examining Fig. 3b, one can see that the difference between the J-N-limit and the achieved sensitivity is large for weak

microwave power, but becomes almost negligible for strong microwave power. At the optimal microwave power -18 dBm, the difference is almost nothing. This suggests that the sensitivity enhancement due to the spin refrigeration effect, is more significant at low microwave power.

The above conclusion can be much clearly seen in the plot of ratio $\Delta\eta/\eta_0$ of the sensitivity reduction $\Delta\eta$ to the sensitivity η_0 . I believe that η_0 is the sensitivity when the system operates in the magnetic insensitive mode. At the microwave power -18 dBm, this ratio is zero, indicating no enhancement. At the microwave power -22 dBm, this ratio is 0.51 dB, indicating a sensitivity enhancement. For lower microwave power, the ratio is about -1dB, indicating a much larger enhancement

From the above discussion, we can see that there is a conflict between the achieved optimal sensitivity and the spin refrigeration-induced enhancement. It would be helpful if the authors could explain clearly under which condition the spin refrigeration effect can enhance the sensitivity, and why they choose -22 dBm as the operating point instead of -18 dBm.

Comment 7: In the equations (2) and (5), there is a period in the end. However, in the equations (5) and (6), the period is missed. There are also similar problems for the equations in the supporting information.

Comment 8: In the second paragraph of Section II B, the text said "To establish a noise baseline, we detune the spin ensemble from the cavity by 5 MHz and measure the noise power spectral density in a 1 Hz band at 15 kHz offset in the signal channel for the bare cavity driven on resonance (blue)". I suggest the authors to add one sentence to relate this description to Fig. 4, and explain why they have chosen 5 kHz offset.

Comment 9: In Fig. 5(b), there is a solid line to distinguish between the normal region and the bistable region. It might be necessary to explain shortly how the biostability occurs and what effect it might have on the sensing performance.

Comment 10: In the second paragraph of Section IV, the authors have discussed about low- ρ , medium- ρ and high- ρ regime. It might be necessary to give some numbers, for example, low- ρ regime ($\rho < xx$ ppm). It might be useful to introduce some vertical lines in Fig. 5(a). There is also a typo for the left axis of Fig. 5(a), and the correct symbol should be Vd/V .

Comment 11: In Section XI B, the authors indicate that they have used their device to measure a DC field around 4 μ T. However, in section IVC of the supporting information, the authors have measured an AC field with amplitude 2 nT. The authors might need to mention the latter measurement in the main text. Besides, from Fig. 4 and Fig. S9(b), we can see that the sensitivity for low frequency is around nT. Would it possible to detect such a small field by averaging the oscillating noise over longer time?

Comments for the supporting information:

Comment 1: Add common or period to the end of expressions when necessary. Mark the equations as "(S1),(S2),...", the figures as "Supplementary Figure S1,...".

Comment 2: In the main text and the supporting information, the symbol $|\alpha|^2$ is interpreted as the cavity occupancy. I am not familiar with the term "cavity occupancy". It seems to me that $|\alpha|^2$ represents the intra-cavity photon number. The authors might need to check this term.

Comment 3: It might be necessary to illustrate the bistability with a picture, and explain how it might affect the sensitivity.

Comment 4: The theoretical treatment in Section II D is very similar to what did in the paper [npj Quantum Information 8, 125 (2022)], see the Supplementary Note 2 of that paper. It might be necessary to cite this paper.

Comment 5: What does the value 1.5×10^7 mean in the end of the section IVA?

Reviewer #2

(Remarks to the Author)

The authors have almost entirely re-written this manuscript in response to both my, and the other reviewers comments. My first response is that the manuscript reads significantly better than the first revision and is much easier to follow. The responses to my comments are very comprehensive with clear changes evident in the manuscript.

I also appreciate the authors emphasizing the novelty of the work specific to this manuscript and I find myself significantly more onboard than I was after reading the first version.

Overall a comprehensive response to my comments and a significant alteration of the manuscript (for the better). I am afraid I have no more useful comments for the authors, and have no issues recommending publication.

Reviewer #3

(Remarks to the Author)

the Authors have addressed and answered all the comments/questions I have raised in the first review. I have no further issues.

I remain in the opinion that this manuscript will fit better in a specialized physics publication, but I find the scientific work and presentation ready for publication.

Reviewer 1 (Remarks to the Author):

In the current manuscript, the authors reported the optimization of the cavity-enhanced microwave readout of the NV ensemble-based magnetometer at room temperature as demonstrated in their previous study [Eisenach, E. R., et al. Nat. Commun., 12(1), 1357 (2021)] by study experimentally and theoretically the influence of the microwave probe and the optical spin polarization. Under the identified optimal condition, the minimal sensitivity is reduced from $3 \text{ pT}/\sqrt{\text{Hz}}$ to $0.58 \text{ pT}/\sqrt{\text{Hz}}$, i.e. an enhancement of five times. During the optimization of their device, the authors identify two interesting phenomena, i.e. microwave power-induced transition of the split reflection spectrum to the single peak reflection spectrum, and the microwave mode cooling. Although these phenomena have been well studied before both experimentally and theoretically, the authors demonstrated here for the first time that they can be used to improve the sensitivity in the CQED-based quantum sensing at room temperature. In the supplementary material, the authors demonstrated even the sensing of an AC magnetic field with an amplitude as low as 5 nT.

The current work presents indeed some new results which deserves the publication. However, by reading the manuscript, I was confused by the presentation of the contents, and have also encountered numerous typos or mistakes, which might degrade the readability of the manuscript. Thus, I would like to suggest the authors to revise systematically their manuscript before publishing it.

We thank the reviewer for the positive comments, particularly their recommendation for publication. We list our revisions point-by-point below. Most importantly, we revised the theoretical model based on the reviewer's suggestion to consider the 8-level NV level structure that fully models the optical spin polarization process as well as spin-lattice relaxation. We are especially impressed with the reviewer's deep understanding and attention to detail.

Comment 1: It seems to me that the title might need to be improved to reflect the sensitivity optimization via the two phenomena mentioned above.

We thank the reviewer for the suggestion. We would like to add some notion of "sensitivity optimization" to the title, but find that this is difficult without making the title too wordy. The concept of optimization is quite general, so simply adding "optimized" seems vague and invites further questions rather than conveying a clear idea. We also tried to add the "within the nonlinear regime," but it also sounds ambiguous about what the nonlinear regime is. We prefer the current title as a compromise between fully detailing the work and simplicity but are open to suggestions or further discussion.

Comment 2: Although the theoretical modelling reproduces very well the experimental results, it is quite difficult for readers to perceive the transition from the linear regime to the nonlinear regime. Can the authors come up with some schemes to illustrate the physics behind?

We thank the reviewer for the comment. The nonlinearity can be understood as any modification of the reflection spectrum *itself* with changing input power—a reflectivity that does not simply relate input to output with a fixed coefficient (linear) but a coefficient that changes with power (nonlinear). This changing spectrum can be seen in Fig. 2b and is demarcated with the label P_s . We have emphasized this intuition and relation in the main text.

Change in the text: we added text to describe the nonlinear regime following Eq. (3) of the revised manuscript:

As $f(\beta_{\text{in}})$ has monotonic, linear dependence with β_{in} , increased probe power will result in increased signal for fixed reflectivity change $\partial \text{Im}[r]/\partial \omega_s$. However, in the limit of large drive power on resonance with both the cavity and the spin ensemble, the spin polarization is suppressed and the ensemble is effectively decoupled from the resonator. This results in a nonlinearity: the reflection coefficient itself changes with input field, meaning β_{out} is no longer linearly related to β_{in} . This effect is shown in Fig. 2a, with full spectra at selected powers shown in Fig. 2b.

Comment 3: It seems that the influence of the microwave cooling is only visible by averaging over tens of measurements. Can the authors provide an explanation for this?

We thank the reviewer for pointing this out. We quantify mode cooling as a reduction in the voltage power spectral density within the cooled band, which can be thought of as the variance of the voltage signal (in the white noise limit). We can only estimate this noise N with some precision given by the standard deviation of the noise measurement with our sample size (confusing because the noise is itself a variance; our precision is the standard deviation of voltage variance). The scaling of the noise measurement precision is ideally $\text{SD}(N) = 2\sigma/\sqrt{n-1}$, with σ is the mean noise value (units of power, V^2) and n is the number of the noise measurements (here we are assuming a constant noise). We therefore need $\sqrt{2(n-1)}$ to be significantly larger than the mode cooling effect to clearly resolve the cooling, for example, resolving a 0.1 dB change in noise power. This would require $0.1 \text{ dB} = 2/\sqrt{n-1}$, or 1250 measurements to reach $\text{SNR} = 1$. We mentioned in the main text that we took 3600 measurements to measure the cooling power of 0.5 dB. This will give an $\text{SNR} \approx 10$ so that we can observe a clear cooling effect.

This approach is also regularly used, for example in Fescenko *et al.* [1], Eisenach *et al.* [2], or Wilcox *et al.* [3].

Comment 4: The authors might need to provide some schematics to illustrate the different kinds of measurement setups, which will be helpful for the explanation of the noise analysis.

We thank the reviewer for the suggestion. We agree that this would improve the clarity of the manuscript. We have added a figure in the Supplementary Materials to illustrate the measurement setup.

Change in the text 1: We have added a figure (Fig. S1) in the Supplementary Materials to illustrate the measurement setup, showing the cavity mode profile and bare cavity measurement. Please also see Fig. 1 of this document.

Change in the text 2: We have added a paragraph in the supplementary materials to discuss the measurements:

We employ a homodyne circuit to measure the phase change induced by the NV-cavity system as shown in Fig. S1(a). Probe microwaves from a signal generator are divided into a reference arm and a signal arm using a Wilkinson microwave power divider. The reference arm is directly connected to the LO port of a mixer (HX3400), with a voltage-controlled phase shifter to tune the relative phase. The signal arm is directed to a circulator with tunable attenuation for power control on the cavity input, and the circulator's

FIG. 1. (a). Experimental setup for cQED sensor. (b). COMSOL simulation for the dielectric resonator. Our dielectric resonator contains two cylinder with outer diameter: 0.669 inch; inner diameter: 0.236 inch; Thickness: 0.314 inch. (c). The cavity measurement with critical coupling. It shows a linewidth of 260 kHz, corresponding to an intrinsic linewidth of $\kappa_c = 130$ kHz. The cavity center frequency is around 2.877 GHz.

microwave output is coupled into the dielectric resonator using a probe loop. The reflected signal from the cavity returns to the circulator and is connected to a low-noise amplifier and subsequently the RF port of the mixer for a homodyne measurement. This setup effectively separates the reflected microwave signal from the incident signal, enabling the measurement of the quadrature part of the reflection coefficient by an appropriate setting of the LO phase shifter. The quadrature signal is subsequently digitized using a sampling rate of 200 kS/s.

The diamond ($3 \text{ mm} \times 3 \text{ mm} \times 0.9 \text{ mm}$, 4 ppm NV ensemble, sourced from Element 6) is set at the $\text{TE}_{01\delta}$ mode maximum point in the center of the dielectric resonator (Skyworks, $\epsilon_r \approx 31$). See Fig. S1(b) and S1(c) for cavity mode simulation and reflection measurements. Note that the diamond size is much smaller than the mode homogeneous region so it is fair to assume a homogeneous single spin coupling strength g_s . A 4H-SiC wafer is used for heat transfer and supporting the diamond, while two pieces of low-loss-tangent polytetrafluoroethylene (PTFE) are used to fix and align the dielectric resonator. An aluminum shield is employed to isolate the system from external signals in the lab, such as WiFi and 3G signals (1.9 GHz), and to reduce radiative losses. An 8W 532 nm pump laser is utilized to optically polarize the spin ensemble. External magnetic bias is provided by 3-axis magnetic coils.

In the noise measurements, we first connect the LNA to a 50Ω resistor to set a baseline for the Johnson-Nyquist limit. Then we reconnect the cavity and gradually changed the tunable attenuator to test the spin refrigeration with different input microwave power.

Comment 5: The manuscript might need to be reconstructed. Since the paper aims to optimize the sensitivity through the microwave probe and the optical polarization, it might be necessary to give the expression for the sensitivity and the signal in the very beginning, and indicates then that the signal should be optimized through the nonlinear model of the reflection coefficient.

We thank the reviewer for the suggestions. We have reconstructed the main text significantly, emphasizing spin refrigeration as a fundamental step for sensing applications and also giving the expression for the sensitivity in the very beginning, as suggested by the reviewer. We have also removed the discussion of linear-regime theory in the main text to improve the readability and clarity of the manuscript by moving to the novel material more quickly. We include this linear-regime theory in the Supplementary Materials.

Change in the text: We have significantly modified the main text, emphasizing sensitivity and associated signals quickly following the introduction. Please see Sec. II in the main text.

Comment 6: It might be necessary to provide the line-cuts for $\Delta_s = 0$ in Figure 2b to illustrate the shape of the imaginary part of the reflection coefficient.

We thank the reviewer for the suggestion and have added another figure in the Supplementary Materials to show these shapes more clearly. We show a linecut for $\Delta = 0$ as spin frequency is changed, which is directly related to the sensitivity, both as a change in reflection coefficient and as a change in measured voltage. We also add a 2D model plot to show the power-frequency dependence of the imaginary part of the reflection coefficient for $\Delta_s = 0$ in Fig. S5. We believe this will not only illustrate the shape of the imaginary part of the reflection coefficient but also give an explicit expression about the saturation behavior for $\Delta_s = 0$.

Change in the text: we have added a figure (Fig. S5) in the Supplementary Materials to show the line-cuts for $\Delta = 0$ and the model plots for $\Delta_s = 0$ with a paragraph to explain the behavior.

Comment 7: The thermally-polarized exceptional point model should be explained in more details. It is not clear what is the difference between this model and the model used in the current study.

We thank the reviewer for mentioning this point. The model presented in Zhang *et al.* [4] solves the semiclassical Maxwell-Bloch equations to determine the ensemble polarization under a possibly-strong microwave drive. They then use this reduced polarization to define a microwave-drive-dependent effective coupling constant between the spin ensemble and the microwave resonator. From this effective model they calculate the microwave spectroscopic signature of the system [their main text Eq. (1)]. In contrast, we solve the semiclassical Maxwell-Bloch equations

FIG. 2. (a) 1D linecut of the imaginary part of the reflection coefficient $\text{Im}(r)$ for $\Delta_s = 0$ for different microwave power P . The parameters used are described at the end of Sec. S1. (b) 1D linecut along $\Delta = 0$ for the voltage output. Yellow: -50 dBm, Blue: -18 dBm, Red: -8 dBm. (c). 1D linecut along $\Delta = 0$ for the spectrum in Fig. 2b in the main text.

self-consistently to determine the cavity field in the combined presence of the microwave drive and the subordinated spin ensemble. We then determine the spectroscopic signature from the cavity field. The additional effect captured by our approach is power broadening of the ensemble linewidth due to the strong cavity microwave field at high drive powers. These approaches lead to qualitatively different scaling of the sensing signal for strong microwave drive, and our treatment correctly predicts the rollover of the sensing signal beyond saturation. Additionally, we consider an inhomogeneous spin ensemble while they consider a homogeneous ensemble.

Change to text: we have added the following paragraph at the end of Sec. IIa

In contrast to previous efforts, this treatment of ensemble inhomogeneity and the optical polarization rate explains the value of the optimal microwave power and provides a better fit for our measured signal S in the nonlinear regime. For comparison, we plot the magnetic sensitivity derived from the model presented in Zhang *et al.* [4] in Fig 2c. This analysis calculates the expected polarization of the spin ensemble under the microwave drive and uses this to construct an effective linear Hamiltonian with a reduced, polarization-dependent coupling constant. This approach captures the quenching of the ensemble-resonator coupling at strong microwave drive, which is the focus of their study, but not the power-broadening of the ensemble due to the resonator field. We self-consistently determine the cavity field in the presence of the microwave drive and the spin ensemble, and then calculate the spectroscopic properties of the system from the cavity field. This accurately models the reduced signal in the strong probe regime and the optimal probe power shown in Fig. 2c.

Comment 8: The optical polarization is indicated sometimes as γ_0 , like Figure 2d and e, but is denoted sometimes as γ_p . The authors should make sure the consistency of the symbols.

We thank the reviewer for pointing this out. We have substantially rewritten the theoretical analysis in the supplement, including consistent use of γ as the single-NV decoherence rate (homogeneous linewidth) and γ_p as the effective optical polarization rate. In response to another of the referee's comments, the effective optical polarization rate is

now derived in terms of the optical excitation rate in Supplementary Materials Sec. IID2.

Change in the context: we have updated the notation and made the aforementioned substantial revisions to the supplemental material.

Comment 9: In section III A, it is said that the sensitivity-maximizing operating point is -22 dBm. However, the discussion in the previous section seems to indicate a power -18 dBm. The authors need to verify the value.

We thank the reviewer for pointing this out. The *signal-maximizing* operating point is -18 dBm, but the *sensitivity-maximizing* operating point is -22 dBm. This apparent discrepancy is due to the mode cooling effect and the signal generator phase noise, both of which cause the noise to depend on the input microwave power. We have clarified this in the text.

Previously it is confusing partially because in the Fig. 2d,e we mark the -18 dBm as the “Operation point” and provide the linecut for this microwave power. To make this point to be more clear, we now mark the -22 dBm as the operational point (sensitivity maximizing operating point) and provide the 1D linecut for this microwave power in Fig. 2e.

Change in the text: we have added clarifying language when discussing the signal response:

We achieve a maximum signal response of 52 nV/Hz at $P = -18$ dBm and $\gamma_p = 2\pi \times 1.36$ kHz (point 2 in Fig. 2c), limited by the output power of our polarizing laser. The optimal-sensitivity operating point considering power-dependent noise effects (discussed below) is at $P = -22$ dBm, indicated by the blue circle in Fig. 2d.

And additional clarification in the discussion of the mode cooling performance:

At the sensitivity-maximizing operating point $P = -22$ dBm (note that this is different from the signal optimal point -18 dBm), we observe a noise power reduction of 0.51 dB, demonstrating that spin refrigeration offers sensitivity enhancement in relevant conditions.

Comment 11: If we assume $g_s = 0$ in the last expression of Eq. (2), we will get an analytical solution to the steady-state population p_j , which decays exponentially to zero. It means that independent of the laser power, all the spins are polarized to the ground state, which does not agree with what is expected. This problem occurs since the authors do not account for the spin-lattice relaxation in a proper way. In some part, the author argued that the intrinsic spin relaxation is ignored because it is smaller than the optical spin polarization rate and the microwave-induced depolarization rate. However, the spin-lattice relaxation amounts to about 200 Hz at room temperature, which is not too small than the minimal optical polarization rate 1kHz. I suggest the authors to check the models in the article [Zhang, Y., et al., Phys. Rev. Lett., 128(25), 253601 (2022)], and model the spin-lattice relaxation more accurately.

We thank the reviewer for pointing this out. We have substantially rewritten the theoretical analysis in supplementary material Sec. II in response to this and related comments. We now derive effective rates for the optically-induced

and thermal transitions between the states in the ground state triplet subspace. We use these rates to derive Maxwell-Bloch equations for the operators describing three-level ground state triplet subspace. The steady-state solutions to the two-level model used in the main text are equivalent to those of the larger model in the limit where the $m_s = -1$ to $m_s = 0$ and $m_s = -1$ to $m_s = 1$ transitions are off resonance with the microwave drive (satisfied in our experiment). The equations of motion in the main text are now written in terms of the equilibrium (in the absence of microwave drive) polarization and associated longitudinal relaxation rate, which are derived from the analysis of the optical polarization cycle and include the effect of spin-lattice relaxation. We also cite the paper the referee mentions in our main text.

Change in the text: we have recast the Maxwell-Bloch equations [main text Eq. (2)] for the system in terms of the equilibrium polarization $\bar{\mathcal{P}}$ and longitudinal relaxation rate γ_p , and we have adjusted the surrounding text accordingly. We have also added a sentence at near the end of Sec. IIB:

Note that for weak optical illumination, the equilibrium polarization $\bar{\mathcal{P}}$ in Eq. (2) is suppressed while the longitudinal relaxation rate γ_p approaches the intrinsic spin-lattice relaxation rate [5] (see Supplementary Materials Sec. II).

Change in the text: we also significantly change the theoretical part for the theoretical part with a generalized Maxwell-Bloch equation model. See Supplementary Materials Sec. IID2.

Comment 12: Below equation (S2c), it is said that p_j is the population on the state $|\pm 1\rangle$. However, later on, it is said that it is the population difference. The authors should check through the manuscript to ensure the consistent interpretation of the same term.

We thank the reviewer for pointing this out. There was a typo in the description of p_j , all the p_j in the theory are the population on state $|1\rangle$. We fixed this typo based on the reviewer's comment. We also define $\mathcal{P}_j = p_{0j} - p_{1j}$ as the polarization—the population difference between the $m_s = 0$ (p_{0j}) and $m_s = 1$ (p_{1j}) populations in the revision.

Change in the text: We fixed those typos based on the reviewer's comment.

Comment 13: Below the equation (S6), it is said that the population is perfectly initialized to $|\pm 0\rangle$ state for $|\alpha| = 0$, under the assumption of large NV polarization rate γ_0 . However, it seems to me that γ_0 is the spin relaxation rate, while γ_p is the optically induced polarization rate.

We thank the reviewer for pointing this out. This was a typo. The relevant text has been revised in response to other referee comments. In the revised version of the manuscript γ_p is the longitudinal relaxation rate, γ is the homogeneous linewidth (including contributions from the optical polarization cycle and thermal relaxation) and γ_0 parameterize additional contributions to transverse relaxation.

Change in the text: we have revised the definitions of the various gammas based on the updated theoretical analysis

and revised the relevant portions of text to clarify the new definitions.

Comment 14: In the equation S10, lots of new terms are introduced, and some are not explained immediately. What is the function of h ? What is Γ ? In addition, there is a large gap between Eq. (S9) and Eq. (S10). The authors need to explain better.

Thank you for pointing this out, we agree that the transition to this analysis was too abrupt and that the clarity of the discussion should be improved. We have rewritten what is now Supplementary Sec. II(B) following Eq. (S8) with an expanded treatment of the relevant algebra, thorough introduction of notation, and expanded discussion. The functions f and h are standardized (i.e. normalized with unit width) lineshapes introduced to allow analysis of the convolution in terms of physically-relevant dimensional constants and integrals of order unity.

Change in the text: We have rewritten what is now Supplementary Sec. IIB to address the above issues.

Comment 15: In the supplementary section H, Eq. (48) is described as the equations for the atomic coherence and the atomic inversion. The authors should change "atomic" by "spin". In addition, p_j in Eq. (48) describes the population inversion, while it is said as the population on the $|\pm 1\rangle$ spin states before. Such typos or mistakes should be corrected.

We thank the reviewer for pointing this out. We have uniformly replace the term 'atomic' with the term 'spin' throughout the manuscript.

Change in the text: we have substituted 'spin' for 'atomic' in the manuscript.

Comment 16: How is the polarization defined in Eq. (55)? For large pumping, it is about $(1 - \eta)/(1 + 2\eta) \approx 0.32$ with $\eta \approx 0.4$. Such a polarization seems to be too small for me. The polarization about 80% is normally reported in literature.

Thank you for pointing this out. We have made major revisions to the model and rewritten the entirety of Supplementary Sec. II to better describe the NV optical polarization cycle and resulting polarization. The definition of polarization used was the difference in population between the $m = 0$ and $m = 1$ state in the 3A_2 manifold:

$$\mathcal{P}_j = p_{0j} - p_{1j}.$$

The estimate presented was based on primarily on parameters from references we consulted at the outset of this work. In response to this referee comment, we have updated the parameters based on those cited in Barry *et al.* [6]. In particular, we have used the parameters measured by Gupta *et al.* [7]. We have also updated the formulation of the rate equation analysis to present the relevant incoherent transition rates in terms of branching ratios (derived from the aforementioned references) and the optical excitation rate. The revised analysis yields a $m_s = 0$ population of approximately 84% at strong optical illumination and for weak microwave drive. This corresponds to a polarization of 76%, which seems to be consistent with expectations.

Change in the text: we have made major revisions to the rate equation analysis and subsequent modeling. This is now presented in Supplementary Sec. IID and connected to the model used in the main text and the beginning of Supplementary Sec. II. We have rewritten the entirety of Supplementary Sec. II.

Comment 17: In the last part of the supplementary material, the different kinds of noise are discussed. It might be helpful if the authors can provide a schematic for the homodyne measurement setup.

We thank the reviewer for pointing this out. We added the schematic plot in the supplementary materials. Please see also see the reply in the Comment 4.

Change in the text: We add a Supplementary figure and related discussions on the schematic for the homodyne measurement setup. We also add a paragraph about the noise measurement:

In the noise measurements, we first replace the cavity with a 50Ω resistor to set a baseline for the Johnson-Nyquist limit. Then we reconnect the cavity and gradually change the tunable attenuator to test the spin refrigeration with different input microwave power.

Comment 18: It seems that the caption for the Supplementary Figure 6 and 7 are mistakenly exchanged. The authors should correct this mistake.

We thank the reviewer for catching this and we have exchanged the captions to their intended figures.

Change in the text: We exchanged the caption for these two figures (Fig. S7 and Fig. S8 in this version).

Reviewer 2 (Remarks to the Author):

I have sectioned off my comments to focus on the aspects highlighted in the reviewers guide for Nature publishing group articles.

We thank the reviewer for the generally positive comments and really well-formatted review. We list our revisions point-by-point below.

Key results

The Authors demonstrate a room temperature magnetic field sensor that utilises an ensemble of NV centres strongly-coupled to a microwave cavity. The Authors spin measurement is based on a readout of microwave photons rather than probing the optical transitions in a more typical ODMR setting. These microwaves are used to directly probe the ground state spin sublevels of the NV ensemble. By doing this they are able to remove the complexity of designing an optical cavity to enhance their measurement sensitivity. Instead, they can use microwave cavities where it is significantly easier engineer a cavity that can reach the strong coupling regime. The notable claim seems to be a record sensitivity for magnetometry using an NV centre, which is an interesting result and one that others will look to use as a reference point as a benchmark for comparison in the future. The results seem to be built on several innovations namely the strongly coupled microwave cavity, the “spin cooling”, and control of the microwave/optical-polarisation power. I have some questions surrounding which of these are new to this manuscript (see clarity and context)

We thank the reviewer for this summary and the insight into what the manuscript is communicating. Our intent is to highlight the innovations noted by the reviewer - the microwave mode cooling effect in an active sensor and a thorough understanding of the nonlinearities and tradeoffs inherent in this type of strongly-coupled cQED sensor. While we do achieve state-of-the-art sensitivity and readout fidelity, this is more to show the potential utility of the innovations at the cutting edge rather than claiming that the sensitivity in itself is the most important part of the work. We have significantly reconstructed our manuscript to clarify the novelty and convey these ideas (see the point-by-point revisions below).

Clarity and context

The big conclusion seems to be that “The demonstrated magnetic sensitivity of $580 \text{ fT}/\sqrt{\text{Hz}}$ sets the standard for the highest-performing continuous operation NV sensor”. To the best of my knowledge this seems to be the case. The nearest being the $900 \text{ fT}/\sqrt{\text{Hz}}$ found in reference [13] Ilja Fescenko, et-al, Phys. Rev. Research 2, 023394 (2020), which is not specifically called out during the concluding remarks of the paper. Further, I think the quoted statement is a little ill defined. For example, the use of “continuous operation” looks to me like a qualifier. Is this setup unequivocally the most sensitive NV-diamond magnetometer? Is there some pulsed operation that is more sensitive? Also, there is no context to this statement, the authors should more explicitly state how their results compare to those of others. I suggest providing a direct quantitative comparison to allow the reader to easily interpret the advances achieved.

We thank the reviewer for highlighting the need for direct quantitative comparison to the state-of-the-art in NV sensing. We have added a discussion and comparison to both the introduction and conclusion/outlook of the paper, and present a review here:

To make this point as clear as possible, also combining the question raised by Reviewer 3, we use the figure of merit of “inverse readout fidelity σ_e ” (ratio between the sensitivity and the spin-projection-limit), together with the sensitivity, to compare our sensor with other NV-based systems. This figure of merit normalizes sensitivity to a standard measure of the potential sensitivity using a particular diamond (number of spins, coherence time, gyromagnetic ratio), and so removes improvements that are due to improved sample properties, control techniques e.g. dynamical decoupling or spin bath driving, or enhanced gyromagnetic ratio. It is also a well-defined standard used in previous works Barry *et al.* [6]. Let’s compare different schemes with our sensor.

The best sensitivities and inverse readout fidelities achieved for spin ensembles in continuous regime with traditional ODMR we observe are:

- (a) Barry *et al.* [8]. This shows an inverse readout fidelity of ≈ 5000 for a spin ensemble size of 5×10^{11} and a sensitivity of $15 \text{ pT}/\sqrt{\text{Hz}}$.
- (b) Schloss *et al.* [9]. This shows an inverse readout fidelity of also ≈ 5000 for a spin ensemble size of 10^{15} . The sensitivity is $50 \text{ pT}/\sqrt{\text{Hz}}$.

The previous work Eisenach *et al.* [2] with a cavity-enhanced readout demonstrates an inverse readout fidelity of ~ 1800 for a spin ensemble size of 2×10^{15} and a sensitivity of $3 \text{ pT}/\sqrt{\text{Hz}}$, which is already a factor of 3 better than the traditional ODMR methods, which is the state-of-the-art inverse readout fidelity for a continuous-wave method as far as we know. However, this method shown in Eisenach *et al.* [2] was considered to be limited by the Johnson-Nyquist limit, with a number of $2 \text{ pT}/\sqrt{\text{Hz}}$ and an inverse readout fidelity of 1200, hindering the significance and further exploration about this method.

As a comparison, we demonstrated an inverse readout fidelity of 380 for a spin ensemble size of 10^{14} with a sensitivity of $0.58 \text{ pT}/\sqrt{\text{Hz}}$. Our sensor is at least one order better than the previous traditional ODMR-based sensor for both inverse readout fidelity and sensitivity. More importantly, we mentioned that based on our model, if we optimize the diamond choice and cavity design, it is possible to reach $3 \text{ fT}/\sqrt{\text{Hz}}$ with an inverse readout fidelity of $\sigma_e = 33$ given the achievable and reasonable diamond parameters in the existing publications. We show these results in Fig. 5 in the main text and Supplementary Materials Sec. V and highlight this point in the introduction and outlook sections.

Fescenko *et al.* [10] (the reference paper the reviewer mentioned above) use a flux concentrator to enhance the system’s effective gyromagnetic ratio. This work is surely valid, but the technique employed is not exclusive with our work, and the gyromagnetic ratio enhancement also lowers the spin-projection floor of the system. In an inverse readout fidelity metric, that result is then ≈ 3000 despite the sensitivity of $900 \text{ fT}/\sqrt{\text{Hz}}$ assuming the similar density-dephasing time product. There are other additional issues with flux concentrators, such as geometry-dependent

flux focusing prevents device response from being tied to fundamental constants (i.e. the magnetic field it measured depends highly on the position of the sensor but not generally related to some basic parameters, like gyromagnetic ratio, in other NV-based measurements). More critically, the concentrated field lines of the flux concentrator are often accompanied by large field gradients, which are likely to hinder the use of long intrinsic dephasing times (See the discussions in Barry *et al.* [11]) and so it is not clear that all “good diamonds” could be used with that technique.

A side note is that although the IR-based continuous wave readout gives an inverse readout fidelity of 65, it is hard to scale with a large number of spins, therefore showing a modest sensitivity number of $30 \text{ pT}/\sqrt{\text{Hz}}$ in the best case [12]. The best quantum limit for this method was claimed at $430 \text{ fT}/\sqrt{\text{Hz}}$, while ours could reach $1.6 \text{ fT}/\sqrt{\text{Hz}}$.

Pulsed techniques also offer advantages over continuous-wave methods. In continuous-wave methods, there is a fundamental competition between optical polarization, spin readout, and coherent precession, which worsens sensitivity limits (pulsed schemes can achieve lower sensitivities in principle, given equivalent diamonds in terms of spin-projection limits). Due to the above reasons, in previous work, people defined the “Optically polarized spin projection limit” if they really wanted to compare with the pulsed schemes (See the Supplementary Materials for Eisenach *et al.* [2]). This “Optically polarized spin projection limit” contains the overhead time in the pulse scheme and the power broadening for the CW scheme to make a fair comparison. It is defined as:

$$\eta = \frac{\sqrt{3}}{g\mu_B} \frac{1}{\sqrt{NT_2^*}} \sqrt{\frac{T_1^{op}}{T_2^*}} \quad (1)$$

with the T_1^{op} is the optical polarization rate. Our sensor reach an “optically polarized inverse readout fidelity” of $\sigma_o = 16$. The best σ_o people achieved is $\sigma_o = 65$ [13]. However, this is with a small NV ensemble, therefore the sensitivity is only $100 \text{ pT}/\sqrt{\text{Hz}}$ in this work. Our work reaches both a good sensitivity and inverse readout simultaneously, meaning that the method itself does have the benefit of large spin ensemble readout.

The best published sensitivity for the pulsed scheme is in [14]. A sensitivity of $900 \text{ fT}/\sqrt{\text{Hz}}$ was realized in this paper. The SQL for their diamond is $0.9 \text{ fT}/\sqrt{\text{Hz}}$. Therefore the inverse readout fidelity is around $\sigma_o = 1000$, compared to our achieved $\sigma_e = 360$ and $\sigma_o = 16$.

The best pulse sensitivity claimed before is a paper posted on arxiv [11] (our Ref. [23]). The authors claim to achieve a broadband sensitivity of $460 \text{ fT}/\sqrt{\text{Hz}}$ with $\sigma_o = 330$. The ensemble size in that work is smaller but much more coherent than what we use, allowing optical methods to achieve high readout fidelities, while the pulsed method enables overall sensitivity to narrowly outperform our device (by enhancing the gyromagnetic ratio using double-quantum transitions, and using P1 center driving to extend coherence). Future work employing pulsed methods in the cavity system, removing the broadening from optical polarization and microwave power, would result in the cavity system outperforming ODMR here. Results that use dynamical decoupling can further improve performance but at the cost of bandwidth (narrowband for their $210 \text{ fT}/\sqrt{\text{Hz}}$).

Change in the text: we have rewritten the abstract and introduction to provide a quantitative comparison to other NV ensemble sensors regarding sensitivity and inverse readout fidelity.

Quantum sensors offer the prospect of device operation at the physical limit of performance [15]. Among the many quantum sensing systems, nitrogen-vacancy (NV) centers in diamond have emerged as a promising platform [6, 16–18] due to favorable attributes including room-temperature spin polarization and readout, atomic-scale size [19–22], and long coherence times [23, 24]. NV-based sensors use resonance spectroscopy of the ground-state spin triplet transition frequencies to infer environmental properties, yielding excellent performance in a variety of sensing modalities including magnetometry [10, 14, 25], electrometry [26–28] and inertial sensing [29, 30], even in extreme regimes [31]. Compared with other sensitive magnetometers such as superconducting quantum interference devices or spin exchange relaxation-free atomic vapors [32–34], NVs can operate in ambient conditions without cryogenics or magnetic shielding, but their sensitivity is limited to pT-level [1, 2, 6, 8]. For spin magnetometers, the standard quantum limit (SQL) of sensitivity is $\eta = 1/\gamma_e\sqrt{NT_2^*}$, where γ_e is the gyromagnetic ratio, T_2^* is the spin dephasing time, and N is the number of spins. There is a tradeoff between N and the ability to measure the spins, which is quantified as the inverse readout fidelity σ_e – the ratio between achieved sensitivity and the SQL [6]. For a single-NV sensor, $\sigma_e \sim 1$ can be achieved but the sensitivity is limited to the nT-level [35]. NV ensemble sensors using continuous optical approaches have significantly increased σ_e to ~ 5000 , but the enhancement from the increased spin number $\sqrt{N} \sim 10^6$ leads to a pT-level sensitivity [6, 8, 9]. Approaching the SQL while maintaining spin ensemble size offers the potential for orders of magnitude in performance improvement.

To overcome the limitations of optical readout, the NV spin resonance may be probed in the microwave domain by coupling the NV ensemble to a microwave cavity mode. This approach eliminates fluorescence shot noise as the dominant systematic noise. The-state-of-the-art [2] reported sensitivity for this approach is 3 pT/ $\sqrt{\text{Hz}}$ but remains a factor of $\sigma_e \sim 1800$ above the SQL. The sensitivity of this approach is limited by the signal-to-noise ratio of the microwave readout, where the signal is upper-bounded by spin saturation effects, and the noise floor is determined by thermal Johnson-Nyquist noise. In this work, we address both of these limitations. We comprehensively model nonlinear spin saturation effects in the strong coupling regime, including spin ensemble inhomogeneity, to optimize and predict sensor response. We further apply recently introduced “spin refrigeration” [36–39] techniques to an actively-probed sensor for the first time, reducing the noise floor below the Johnson-Nyquist limit. As a result, we achieve a sensitivity of 576 ± 6 fT/ $\sqrt{\text{Hz}}$ with an inverse readout fidelity of $\sigma_e \sim 360$, which sets the state-of-the-art for a continuous-wave NV magnetometer.

Beyond the present experimental results, our model allows us to outline sensitivity regimes for varying spin ensemble properties, cavity designs, and power requirements. We argue for the feasibility of 3 fT/ $\sqrt{\text{Hz}}$ sensitivity in an optimized device with a $\sigma_e \sim 33$. The demonstration of spin refrigeration in an operating sensor shows the potential for improvement of any Johnson-Nyquist limited system, e.g. micro-electromechanical systems.

We also add a comparison sentence with the pulsed techniques, and the techniques with lock-in detection, which limits the measurement bandwidth.

Next we examine the spectral performance of the device under optimal probe power. In Fig. 4, we plot the magnetic sensitivity in blue across a spectral range of 10 Hz - 100 kHz. The mean sensitivity of a 1 Hz band between 10-11 kHz offset is $\eta = 576 \text{ fT}/\sqrt{\text{Hz}}$ with a standard deviation of $6 \text{ fT}/\sqrt{\text{Hz}}$, and does not roll off at high frequencies within the measurement bandwidth of 100 kHz. This is in contrast to sensors using e.g. dynamical decoupling techniques [6] or with a limited lock-in bandwidth [16] that have significantly reduced spectral response.

The work presented in this manuscript seems to build on previous publications specifically: reference [24] E.R.Eisenach, et al, Nature Communications 12,1357 (2021), and reference [25] D.P. Fahey, K. et al, Phys. Rev. Appl. 20, 014033 (2023). Reference [25] is clearly from the same group of academics, and I think the same for [24] although this is less clear. I think the authors should be more explicit and take ownership of their previous work just to make it clear what they themselves have done previously. This is problematic particularly in the abstract where they state “Here we move towards this quantum limit... by introducing (i) a cavity quantum ... (ii) a comprehensive nonlinear model... (iii) spin refrigeration”. (i) was introduced in ref [24] and (iii) in [25]. I think the authors should put more focus on the specific innovations present in this manuscript whilst acknowledging how it builds on their previous efforts. My understanding is the main innovation is due to the control and modelling of the MW power/optical spin-polarisation rate by moving into the non-linear regime? I think this could be communicated more effectively with the reader.

We thank the reviewer for pointing out that the abstract and text could be more clear in differentiating this work from previous efforts, in particular how our work overcomes existing issues in cavity readout by the application of mode cooling and understanding of nonlinear effects, and extends the concept of mode cooling to an actively-probed sensor.

Change in the text: we have rewritten the abstract as follows:

Quantum sensors based on solid-state defects, in particular nitrogen-vacancy (NV) centers in diamond, enable precise measurement of magnetic fields, temperature, rotation, and electric fields. Cavity quantum electrodynamic (cQED) readout, in which an NV ensemble is hybridized with a microwave mode, can overcome limitations in optical spin detection and has resulted in leading magnetic sensitivities at the pT-level. This approach, however, remains far from the intrinsic spin-projection noise limit due to thermal Johnson-Nyquist noise and spin saturation effects. Here we tackle these challenges by combining recently demonstrated “spin refrigeration” techniques with comprehensive nonlinear modeling of the cQED sensor operation. We demonstrate that the optically-polarized NV ensemble simultaneously provides magnetic sensitivity and acts as heat sink for the deleterious thermal microwave noise background, even when actively probed by a microwave field. Optimizing the NV-cQED system, we demonstrate a broadband sensitivity of $576 \pm 6 \text{ fT}/\sqrt{\text{Hz}}$ around 15 kHz in ambient conditions. We then discuss the implications of this approach for design of future magnetometers, including near-projection-limited devices approaching $3 \text{ fT}/\sqrt{\text{Hz}}$ sensitivity enabled by spin refrigeration.

We have also rewritten the introduction, please refer to the comments above. We also highlight that the spin

refrigeration persists in an actively-driven system:

At the sensitivity-maximizing operating point $P = -22$ dBm (note that this is different from the signal optimal point -18 dBm), we observe a noise power reduction of 0.51 dB, demonstrating that spin refrigeration offers sensitivity enhancement for an actively-driven system until phase noise dominates the noise performance or spin saturation effects reduce the spin-refrigeration behaviour.

It needs to be made more explicit what new innovations specific in this manuscript have led to the increased sensitivity. I also think the authors should place the results in wider context by comparing them with other magnetic sensors including SQUID and atomic based. I appreciate comparisons can be tricky and each implementation will have unique advantages and disadvantages. However, as it stands the non-specialist reader will not be able to interpret the significance of the implementation in this manuscript.

We appreciate the constructive feedback and have modified the manuscript to more clearly and efficiently communicate the novelty and main innovations of the work, in particular how it differs from previous work. In addition we have added a comparison to other magnetic sensors. To make the novelty more clear, we add additional discussion on the limitations of NV-cavity sensors and emphasize how our approach addresses these issues in a new way.

Change in the text: we have added the following discussion to the introduction. Please see the comments above.

Change in the text: we have added the following sentence in the first paragraph of main text Sec. I to address the advantage of NV systems compared to other magnetometers. It is also the motivation of our work to try to make the sensitivity number for the NV sensor to be competitive with those techniques.

Compared with other sensitive magnetometers such as superconducting quantum interference devices or spin exchange relaxation-free atomic vapors [32–34], NVs can operate in ambient conditions without cryogenics or magnetic shielding, but their sensitivity is limited to pT-level.

More specific clarifications:

Why does changing the density of NV centres not impact the sensitivity whereas the volume of the Diamond does? This doesn't make intuitive sense to me and I would appreciate some clarification.

In our modeling of density, we have assumed that NV density trades off linearly with both inhomogeneous and homogeneous spin dephasing: $\Gamma = a\rho$ with $a = 2\pi \times 82.5$ kHz/ppm and $\gamma_0 = b\rho$ with $b = 2\pi \times 7.5$ kHz/ppm based on our current device, following Bauch *et al.* [40], due to inherent dipole-dipole interactions. As a result, increasing density will increase the ensemble coupling g as $\sqrt{\rho}$ (g^2 as ρ), but potentially at the cost of increased spectral widths Γ, γ depending on the relative contribution of spin-spin interactions compared to other effects (mainly laser-induced dephasing).

Change to text: we have added the following intuition for the density-dependent sensitivity:

In the low- ρ regime, laser-induced dephasing dominates both the inhomogeneous and homogeneous NV linewidth; therefore, adding more density to the system increases the collective coupling without the cost

of broadening the NV spins. In the medium- ρ regime, the density-induced inhomogeneous broadening eclipses the laser-induced dephasing, and the sensitivity is almost independent of density ρ . In the high- ρ regime, the spin dipole-dipole homogeneous dephasing starts dominating the laser-induced dephasing, causing a sensitivity drop in this domain.

Section III, B related to broadband operation does not quantify what counts as broadband. This could be remedied via comparison with other results in both diamond colour centres and indeed other materials/systems. Once again someone who is not familiar with sensing applications will have no context for the claim of broadband operation.

We agree that the meaning of the term “broadband” is ambiguous and may change depending on context. The spectral response of our measurement is quantitatively displayed in Figure 4, with a bandwidth of 100 kHz. The class of sensors we are specifically attempting to differentiate are NV sensors that employ pulsed dynamical decoupling techniques, which can extend spin coherence times (improving sensitivity) at the cost of restricted bandwidth. We have made this more clear by explicitly noting the bandwidth and how it relates to other methods.

Change to the text:

Next we examine the spectral performance of the device under optimal probe power. In Fig. 4, we plot the magnetic sensitivity in blue across a spectral range of 10 Hz - 100 kHz. The mean sensitivity of a 1 Hz band between 10-11 kHz offset is $\eta = 576 \text{ fT}/\sqrt{\text{Hz}}$ with a standard deviation of $6 \text{ fT}/\sqrt{\text{Hz}}$, and does not roll off at high frequencies within the measurement bandwidth of 100 kHz. This is in contrast to sensors using e.g. dynamical decoupling techniques [6] or with a limited lock-in bandwidth [16] that have significantly reduced spectral response.

Validity

The analysis of the results all looks correct as far as I can see. The headline claim of a sensitivity of $580 \text{ fT}/\sqrt{\text{Hz}}$ is backed up by the data. Although it is unclear why the result is stated to this level of precision, the data in Fig.4 looks a little noisy. I would say it is $\approx 600 \text{ fT}/\sqrt{\text{Hz}}$ based on the data in Fig.4. Perhaps some error on this sensitivity could be included?

We thank the reviewer for the comment. We compute the mean power spectral density by taking a series (N=3600) of 1-second measurements and averaging the power in each 1 Hz band, and then convert this to a magnetic field (amplitude) spectral density. The resulting amplitude spectral density has a mean value of $576 \text{ fT}/\sqrt{\text{Hz}}$ in the band from 10 kHz to 11 kHz, with a standard deviation of $6 \text{ fT}/\sqrt{\text{Hz}}$ between the 1 Hz bins in that band. We have added this error to the text.

We describe the sensitivity as follows: The mean sensitivity of a 1 Hz band between 10-11 kHz offset is $\eta = 576 \text{ fT}/\sqrt{\text{Hz}}$ with a standard deviation of 6 fT ,

Significance

I am left a little unclear on the significance of what has been achieved in this manuscript due to lack of clarity on the main innovations that are unique to this manuscript, rather than those in previous publications (refs [24], [25], see also clarity and context)

I do think the headline result for the sensitivity quoted in this manuscript will be of interest to researchers in the field of sensing with defect centres in diamond. It will undoubtedly function as a benchmark that will be used for others working in the field. I think there is the potential for interest from non-specialist readers, however the way the work is currently presented in this manuscript will dampen the appeal (see clarity and context, suggested improvements).

We thank the reviewer for their critical remarks here. We have clarified that this work demonstrates a sensor that takes advantage of mode cooling, unlike previous work that either demonstrated mode cooling phenomena but without any sensor response (active probe), or other work showing sensors without mode cooling. We hope that the application of this effect, which has not to our knowledge been used in any sensor prior to this work, is of broad interest to non-specialists who might seek to apply this mode cooling in their own fields of expertise. It has not been previously shown that mode cooling is compatible with an active probe in contrast to a reduction of passively-monitored Johnson noise; this work shows that the technique does have applications in relevant fields (such as MEMS devices or atomic sensors). We have rewritten the introduction and background to emphasize this more clearly.

Data and methodology

The reflection spectra in Fig 1.c looks very similar to the one in ref [25] apart from the fact the y-axis has been inverted. Since the authors have not properly acknowledged their previous work it is not possible to know if this data is new to this manuscript, i.e., is it the same sample, same experimental conditions, if it is different how is it different? Incidentally similar data is presented in Fig.2 b (for different MW powers) where the y-axis is inverted back to align with the convention in their previous work. A consistent presentation format for the data would be better.

We thank the reviewer for pointing this out. The experiment in ref [25] was done in the Army Research Laboratory, while this work was done at MIT. It is from different NV samples, different cavity designs (the center frequency of ref [25] is 2.895 GHz, while this manuscript is around 2.877 GHz), experimental setups, and conditions. We have added details, including the cavity material and cavity mode simulations, in the Supplementary Materials. It was a mistake on the y-axis for the Fig. 2b. We fixed the problems for a consistent presentation format for the figures like the reviewer mentions.

Change in the text: We add the details about the experimental schematic and descriptions, and we fixed the typos on the y-axis for Fig. 2b.

Analytical approach

This all looks sound the authors are using fairly standard CQED formalism and are using techniques established in previous publications.

We appreciate that the reviewer finds the approach sound and the formalism consistent with the literature.

Suggested improvements

I do not find the energy level diagram in figure 1b particularly illuminating. What is happening between the left- and right-hand side of this diagram? is it illustrating the effect of hyperfine coupling with the ^{14}N ? I think the experiment focusses on transitions between the 0 and +1 ground state manifolds (ignoring the -1), but I can't find where this is explicitly stated. Also, the little balls with arrows that are intended to represent spins (I presume) all point in the same direction is this correct/intentional? If so it is maybe not a good way to differentiate between the transitions. The curved grey lines I think represent the cavity, but these are drawn around a specific transition, implying the others do not couple, which I don't think it the authors intention. The caption for this also needs more information to help clarify this sub-figure.

We thank the reviewer for their comment on Fig. 1b, the energy level diagram. It does emphasize the hyperfine interactions which aren't the main thrust of the paper. We have re-drawn the figure to focus on the coupling of the NV to the cavity and indicate which parameters are being changed in the experiment.

Change in the text: We have rewritten the caption for the Fig. 1b. We have also redrawn Fig. 1b to highlight more about moving the transitions one by one to couple with the cavity and removed the spin icons to avoid confusion.

In general I find the manuscript a little uninspiring. It reads more like thesis chapters. It is quite disjointed and does not flow nicely. I was frequently bogged down in details and found it challenging to follow the main thread of the paper. This makes it even more challenging to unpick the innovations and significance of the result. As I have said the authors should highlight the main innovation specific to this paper that has enabled their result and build the manuscript around this. The reader should be provided sufficient detail to understand the result/innovation. I encourage the authors to consider what information is essential for the reader to understand and interpret the results. If it is not absolutely necessary then it can be moved into the supplement. Also if there are details in previous manuscripts it will be sufficient to summarise the main findings to the reader and redirect them to the relevant publications for further details.

Further issues can be solved by spending more time placing the results in context, to better engage the reader. I think the main innovation here is a detailed study of how controlling the microwave/optical power can increase the sensitivity. The paper jumps immediately into the theoretical CQED background which I find quite jarring. A lot of the content in section II is covered in previous references and much of it looks like standard CQED theory. The non-linear part looks like the main innovation. I wonder if there is a better way to discuss this that focusses more on the results and shows how this is backed up by a detailed theoretical model. In this case much of the CQED theory can be moved to the supplement.

We appreciate the reviewer's suggestions. We have majorly reorganized the manuscript and removed most of the treatment of cQED linear response, which is more-or-less standard. This material is discussed in the Supplementary Materials. We have also reconstructed the manuscript and added additional discussion to place the results in context.

We hope this makes the manuscript both more readable and compelling.

Change in the text: We remove the most of the treatment of cQED linear response, and reconstructed the manuscript. (Please also see the response for the previous comments.)

References

The references section is fairly comprehensive. My only suggestions would be to include extra in response to areas I have highlighted where the manuscript lacks context.

We have added the references associated with the added context.

Change in the text: We have added the references for SQUID and spin exchange relaxation free magnetometers. (Ref. [21-23])

My expertise

Quantum photonics with quantum emitters, spin-photon interfaces, single photon sources, cQED and open quantum systems and quantum technology in general. Familiar with magnetometry using NV centres in diamond, but not a specialist in this area.

Summary

Overall I am somewhat torn for my recommendation for this paper. On the one hand the sensitivity achieved will act as a benchmark. However, this will likely be surpassed by others, including the authors at some point. However, I do acknowledge it is important to have these results published as a reference point. I will leave this as a decision for the editors.

The main problem is the way the manuscript is written is uninspiring and not typical of what I would read in nature communications. I think some significant re-writing will be necessary to improve the accessibility of the manuscript such that it can have the greatest possible impact.

We thank the reviewer again for the insightful suggestions. We have rewritten and reconstructed the manuscript to address accessibility and context, and hope that the reviewer finds it much improved.

We agree with the reviewer that the sensitivity and inverse readout fidelity will act as a benchmark for the NV sensing area. On the concern that the work will be eventually surpassed in sensitivity and therefore made irrelevant or obsolete, we would like to make three points. First is that we have introduced mode cooling to NV sensors for the first time, and continued work optimizing that technique will benchmark and reference to this work. Second is that we have provided a comprehensive model and parameter-space exploration for cQED sensors. As engineering continues to improve and reach e.g. high Q factors or diamond spin coherence times, this work will remain relevant as it has mapped the limits of what one may achieve. Finally, that this direction is likely to be improved upon indicates that this is an active area of research rather than one of little interest. Indeed we hope this work prompts others to

improve upon it, both in experiment and theory.

Reviewer 3 (Remarks to the Author):

General

- the authors use an RF resonator and MW detection setup to read the spin states of NV- centers in diamond in the RF domain (rather than in the optical domain as is commonly done). using this approach, they claim superior readout of the spin states, reaching high sensitivity to changes in the magnetic field down to $f\Gamma/\sqrt{\text{Hz}}$.
- the authors use the power-dependent behavior in the strong-coupling regime to balance the readout photon numbers with the sharpness of spectral features to achieve optimal readout efficiency.
- the scientific methods seem adequate.

We agree with the reviewer's summary of our work, and we answer the reviewer's specific questions and list our revision point-by-point below.

• the authors use two interesting methods: the first being the mode cooling shown by same authors in previous paper (Fahey, D. P., Jacobs, K., Turner, M. J., Choi, H., Hoffman, J. E., Englund, D., & Trusheim, M. E. (2023). Steady-State Microwave Mode Cooling with a Diamond N-V Ensemble. PHYSICAL REVIEW APPLIED, 20, 14033. <https://doi.org/10.1103/PhysRevApplied.20.014033>) and second is the optimization of the MW and laser power to achieve optimal operation of the 'device'. however, this seem to be a small improvement compared with the results described in their previous work, and stands on well-known base, notably the power-broadening of NV centers, and nonlinearity of such experimental schemes (such as cited in the main text) therefore I am not convinced it is of great importance and significance.

We thank the reviewer for pointing out that while the methods used are of interest, how they are different from previous work was not being effectively communicated. We made significant changes to the manuscript to emphasize points of novelty and their significance.

NV sensors based on ODMR have been developed for decades, and people have tried different ways to optimize the sensitivity. However, poor optical collection and the nonperfect branching ratio are problems preventing reaching the spin-projection limit. The best continuous ODMR measurement remains an inverse readout fidelity of ≈ 5000 (Barry *et al.* [8], Schloss *et al.* [9]). We use the inverse readout fidelity to compare methods demonstrated using diamonds of disparate quality, see discussions below. To overcome this problem, the previous work (Eisenach *et al.* [2]) develops an MW cavity-based readout scheme to show an inverse readout fidelity of ≈ 1800 and a sensitivity of $3 \text{ pT}/\sqrt{\text{Hz}}$, which was a step towards the spin-projection limit and introduced this exciting method, but remained on a similar performance level with optical techniques. Critically, this method shown in Eisenach *et al.* [2] was considered to be limited by the Johnson-Nyquist limit, with a number of $2 \text{ pT}/\sqrt{\text{Hz}}$ and an inverse readout fidelity of 1200, hindering the significance and further exploration about this method.

We overcome this limitation this by (1) using the spin-refrigeration effect to surpass the room temperature Johnson-Nyquist limit. (2) developing a more comprehensive model to understand and optimize the NV-cavity parameters. We successfully reach an inverse readout fidelity of 360 and a sensitivity of $580 \text{ fT}/\sqrt{\text{Hz}}$. Based on our model, we also point the way to even go to an inverse readout fidelity of 33 and a sensitivity of $3 \text{ fT}/\sqrt{\text{Hz}}$ with achievable diamond/cavity choices by this method. This advance in fully understanding, modeling, and measuring the cQED

systems now clearly shows the potential advantage for this approach in comparison to standard optical techniques.

We note that compared with our previous work, Fahey *et al.* [36], (1) To take advantage of this technique in sensing applications is not straightforward because the microwave probe tone brings the system out of linear response. Our previous work had no input power to the cavity while operating as a refrigerator; this work quantifies mode cooling operation with active driving and the resulting contribution to sensitivity. (2). We comment that this theory does not just involve the power-broadening of NV centers that people have developed before, for example, in the continuous-wave ODMR case. Previous approaches have not simultaneously captured the nonlinear interplay between spin polarization, broadening, cavity-coupling, nor the impact of inhomogeneity. For example, we compare our treatment to that of Zhang *et al.* [4] in the main text and show that our model gives a better fitting and prediction of the experiment.

To clarify this point, we made a thorough reconstruction of the main text, highlighting the main innovations. We appreciate the reviewer for identifying these areas where we could be more clear in the significance and novelty of the work.

Change in the text: We made major reconstructions of the main text, in particular the introduction and abstract, to make it more clear about the significance and novelty.

Quantum sensors offer the prospect of device operation at the physical limit of performance [15]. Among the many quantum sensing systems, nitrogen-vacancy (NV) centers in diamond have emerged as a promising platform [6, 16–18] due to favorable attributes including room-temperature spin polarization and readout, atomic-scale size [19–22], and long coherence times [23, 24]. NV-based sensors use resonance spectroscopy of the ground-state spin triplet transition frequencies to infer environmental properties, yielding excellent performance in a variety of sensing modalities including magnetometry [10, 14, 25], electrometry [26–28] and inertial sensing [29, 30], even in extreme regimes [31]. Compared with other sensitive magnetometers such as superconducting quantum interference devices or spin exchange relaxation-free atomic vapors [32–34], NVs can operate in ambient conditions without cryogenics or magnetic shielding, but their sensitivity is limited to pT-level [1, 2, 6, 8]. For spin magnetometers, the standard quantum limit (SQL) of sensitivity is $\eta = 1/\gamma_e\sqrt{NT_2^*}$, where γ_e is the gyromagnetic ratio, T_2^* is the spin dephasing time, and N is the number of spins. There is a tradeoff between N and the ability to measure the spins, which is quantified as the inverse readout fidelity σ_e – the ratio between achieved sensitivity and the SQL [6]. For a single-NV sensor, $\sigma_e \sim 1$ can be achieved but the sensitivity is limited to the nT-level [35]. NV ensemble sensors using continuous optical approaches have significantly increased σ_e to ~ 5000 , but the enhancement from the increased spin number $\sqrt{N} \sim 10^6$ leads to a pT-level sensitivity [6, 8, 9]. Approaching the SQL while maintaining spin ensemble size offers the potential for orders of magnitude in performance improvement.

To overcome the limitations of optical readout, the NV spin resonance may be probed in the microwave domain by coupling the NV ensemble to a microwave cavity mode. This approach eliminates fluorescence shot noise as the dominant systematic noise. The-state-of-the-art [2] reported sensitivity for this approach is $3 \text{ pT}/\sqrt{\text{Hz}}$ but remains a factor of $\sigma_e \sim 1800$ above the SQL. The sensitivity of this approach is limited

by the signal-to-noise ratio of the microwave readout, where the signal is upper-bounded by spin saturation effects, and the noise floor is determined by thermal Johnson-Nyquist noise. In this work, we address both of these limitations. We comprehensively model nonlinear spin saturation effects in the strong coupling regime, including spin ensemble inhomogeneity, to optimize and predict sensor response. We further apply recently introduced “spin refrigeration” [36–39] techniques to an actively-probed sensor for the first time, reducing the noise floor below the Johnson-Nyquist limit. As a result, we achieve a sensitivity of 576 ± 6 fT/ $\sqrt{\text{Hz}}$ with an inverse readout fidelity of $\sigma_e \sim 360$, which sets the state-of-the-art for a continuous-wave NV magnetometer.

Beyond the present experimental results, our model allows us to outline sensitivity regimes for varying spin ensemble properties, cavity designs, and power requirements. We argue for the feasibility of 3 fT/ $\sqrt{\text{Hz}}$ sensitivity in an optimized device with a $\sigma_e \sim 33$. The demonstration of spin refrigeration in an operating sensor shows the potential for improvement of any Johnson-Nyquist limited system, e.g. micro-electromechanical systems.

We also modify the abstract to be more clear about the novelty:

Quantum sensors based on solid-state defects, in particular nitrogen-vacancy (NV) centers in diamond, enable precise measurement of magnetic fields, temperature, rotation, and electric fields. Cavity quantum electrodynamic (cQED) readout, in which an NV ensemble is hybridized with a microwave mode, can overcome limitations in optical spin detection and has resulted in leading magnetic sensitivities at the pT-level. This approach, however, remains far from the intrinsic spin-projection noise limit due to thermal Johnson-Nyquist noise and spin saturation effects. Here we tackle these challenges by combining recently demonstrated “spin refrigeration” techniques with comprehensive nonlinear modeling of the cQED sensor operation. We demonstrate that the optically-polarized NV ensemble simultaneously provides magnetic sensitivity and acts as heat sink for the deleterious thermal microwave noise background, even when actively probed by a microwave field. Optimizing the NV-cQED system, we demonstrate a broadband sensitivity of 576 ± 6 fT/ $\sqrt{\text{Hz}}$ around 15 kHz in ambient conditions. We then discuss the implications of this approach for design of future magnetometers, including near-projection-limited devices approaching 3 fT/ $\sqrt{\text{Hz}}$ sensitivity enabled by spin refrigeration.

We add a concluding sentence to highlight that the spin refrigeration persists in an actively-driven system:

At the sensitivity-maximizing operating point $P = -22$ dBm (note that this is different from the signal optimal point -18 dBm), we observe a noise power reduction of 0.51 dB, demonstrating that spin refrigeration offers sensitivity enhancement for an actively-driven system until phase noise dominates the noise performance or spin saturation effects reduce the spin-refrigeration behaviour.

- as a ‘device’ or ‘sensor’ author statements seems somewhat ‘overselling’: the authors use a 8W green laser to pump the system, and still need hours-long measurements to achieve the claimed sensitivity, and need specialized electronics, so while the sensitivity is impressive, it is not quite a ‘device’ at this point.

We thank the reviewer for pointing this out. The system does use an 8 W laser, but it does not need hours-long measurements to achieve the claimed sensitivity. The figure of merit, sensitivity, is a measure of how quickly a sensor can resolve a test field from the intrinsic noise background. By convention, it is defined as the minimum detectable field within a 1 Hz bandwidth. Accordingly, our PSDs are generated by measurements with 1 second durations with a sampling rate of 200 kS/s. The 1 second measurement duration is set by convention and the sensitivity we report is a measure of the high quality of our NV-cavity system. In order to accurately determine the variation of these measurements (reduce error in the sensitivity estimate) we must take many measurements over the course of an hour. To clarify this point, we add a figure in Supplementary Materials and corresponding explanations.

On the choice of wording including “device” and “sensor”, one dividing line is whether the system in question is developed for a purpose or not. So a single NV center in diamond whose properties are unknown is likely not a “device”, but a system intentionally assembled for the purpose of magnetometry would indeed be a “device” or “sensor”. It is not an ideal sensor in many aspects, including portability, power consumption, user-friendliness, and so on, but it is still a sensor/device rather than a “sample,” for example. If there is a preferred synonym we can use it; we use sensor/device/system interchangeably to refer to the cQED setup and avoid repetition. We note that previous work has referred to similar laboratory-scale systems as “sensors” (or more specifically as “magnetometers” or “gradiometers”, subclasses of sensor) both in the NV community as well as atomic vapor cells such as [41–43].

Change in the text: We added a figure for the setup and explanations for the sensor’s details in the supplementary materials.

- magnetic field sensitivity: works in optically excited NV centers easily reach sensitivities around 1nT, but this is done with optical powers circa 1mW and on ensembles of <10 NV centers; and potentially with sub-micron spatial resolution. In the system proposed here authors show averaging on a huge ensemble of NV centers and high laser powers, with 3 orders of magnitude increase in accuracy, but loss of spatial resolution. Therefore, I ask to see a figure-of-merit combining parameters such as number of photons in the system and their lifetime(optical and MW), number of NV centers, measurement time, etc, to know if there is really an improvement or is it merely taking advantage of averaging on a huge ensemble.

We thank the reviewer for pointing this out. This sensor does employ an ensemble of spins with a loss of spatial resolution and requires \approx Watt levels of laser power, which makes a comparison to a single-NV device not particularly apt.

The question is what figure of merit is most appropriate—a total sensitivity or a volume-normalized, spin-number-normalized sensitivity, or one including factors like characteristic length scale or power consumption. Neither one is a “real improvement” vs. “merely taking advantage” as characterized here. As a general rule, very good (in many cases ideal) measurements can be performed on small numbers of NV centers, but these techniques do not scale to the level of 10^{14} centers as we have here. In traditional optically detected magnetic resonance (ODMR), both fluorescence collection and optical pumping (needed at high rates near the optical lifetime) are challenging as the sample size is increased. This is exactly the motivation for our work to use a microwave-only readout scheme for large spin ensembles. To clarify this point, we have added an additional discussion of this point in the introduction.

Here the challenge is maximizing the number of NVs that can be read-out at some mean fidelity, which is the total sensitivity metric we have emphasized. For completeness, we have added additional discussion of the spin-projection limit and readout fidelity of our technique, which puts the total sensitivity into a spin-number-normalized context. Here we use the inverse readout fidelity defined as sensitivity/spin-projection limit, which is a standard figure of merit in this field (See for example Barry *et al.* [6]). The state-of-the-art NV ensemble sensors based on continuous ODMR for a spin ensemble are:

1. Barry *et al.* [8] This shows an inverse readout fidelity of ≈ 5000 for a spin ensemble size of 5×10^{11} and a sensitivity of $15 \text{ pT}/\sqrt{\text{Hz}}$.
2. Schloss *et al.* [9] This shows an inverse readout fidelity of also ≈ 5000 for a spin ensemble size of 10^{15} (Note that this number is the total number of spins, but not the polarized spin number). The sensitivity is $50 \text{ pT}/\sqrt{\text{Hz}}$. Laser power 3.3W.

As a comparison, we demonstrated an inverse readout fidelity of 380 for a spin ensemble size of 10^{14} with a sensitivity of $0.58 \text{ pT}/\sqrt{\text{Hz}}$. Our sensor is one order better than the previous traditional ODMR-based sensor for both inverse readout fidelity and sensitivity. We also want to highlight that in the Supplementary Materials, we mentioned based on our model, if we optimize the diamond choice and cavity design, it should be possible to reach $3 \text{ fT}/\sqrt{\text{Hz}}$ with an inverse readout fidelity of 33 given the achievable and reasonable diamond parameters in the existing publications.

To further clarify this point, we add a paragraph in the Supplementary Materials in the sensitivity outlook section. We also add a sentence in the introduction part to highlight the comparison.

Change in the text: We changed the abstract and the introduction to include the inverse readout fidelity discussion and tradeoff for the single NV and spin ensemble NVs to make this point clearer.

For spin magnetometers, the standard quantum limit (SQL) of sensitivity is $\eta = 1/\gamma_e \sqrt{NT_2^*}$, where γ_e is the gyromagnetic ratio, T_2^* is the spin dephasing time, and N is the number of spins. There is a tradeoff between N and the ability to measure the spins, which is quantified as the inverse readout fidelity σ_e – the ratio between achieved sensitivity and the SQL [6]. For a single-NV sensor, $\sigma_e \sim 1$ can be achieved but the sensitivity is limited to the nT-level [35]. NV ensemble sensors using continuous optical approaches have significantly increased σ_e to ~ 5000 , but the enhancement from the increased spin number $\sqrt{N} \sim 10^6$ leads to a pT-level sensitivity [6, 8, 9]. Approaching the SQL while maintaining spin ensemble size offers the potential for orders of magnitude in performance improvement.

- highly specific paper, perhaps more fitting to an applied or basic physics journal.

As noted in this comment, we feel that the work is relevant to both applied and basic physics audiences as we introduce comprehensive cQED theoretical modeling, apply a new noise reduction technique (spin refrigeration), achieve state-of-the-art experimental performance, and show an outlook relevant for potential applications. A focus on a subset of these advances could be more appropriate for a narrow audience, but we hope that their combination (and the widespread applicability of refrigeration for sensing) will make the work of broad interest.

Specific comments.

- the claim of ‘first strong coupling in room temp’ (lines 165-166) is false: it was achieved previously by the same authors in the paper mentioned above.

We thank the reviewer for pointing this out. We mentioned that it is the first for a room-temperature sensor. Previous work in sensing by Eisenach *et al.* [2] did not achieve strong coupling, while mode cooling results from Fahey *et al.* [36] did not measure sensitivity and the associated interplay with the MW drive. However, the achievement of strong coupling is not the main thrust of the paper and could be distracting as here, so we have removed mentions of “first strong coupling” from the text as part of the reorganization of the manuscript.

Change in the text: We remove mentions of “first strong coupling” from the text as part of the reorganization of the manuscript.

- the text should explicitly declare the type (model) and exact design of the cavity, and include a measurements of the cavities’s RF response (can be in supplementary).

We thank the reviewer for pointing this out. We add a section in the Supplementary Materials to describe the cavity details.

Change in the text: We add a figure in the supplementary material to add the COMSOL simulation for the cavity mode profile with the cavity parameters we use in Fig. S1(b). We also add a measurement for pure cavity RF response in Fig. S1(c).

- a sketch of the experimental setup (including MW circuitry and instrumentation) is needed in the supplementary information.

We thank the reviewer for pointing this out. We add a section in the Supplementary Materials to describe the experimental setup, including MW circuitry and instrumentation.

Change in the text 1: We add a figure [Fig. S1(a)] in the Supplementary Materials:

Change in the text 2: We add a paragraph to discuss the measurements:

We employ a homodyne circuit to measure the phase change induced by the NV-cavity system. Probe microwaves from a signal generator are divided into a reference arm and a signal arm using a Wilkinson microwave power divider. The reference arm is directly connected to the LO port of a mixer (HX3400), with a voltage-controlled phase shifter to tune the relative phase. The signal arm is directed to a circulator with tunable attenuation for power control on the cavity input, and the circulator’s microwave output is coupled into the dielectric resonator using a probe loop. The reflected signal from the cavity returns to the circulator and is connected to a low-noise amplifier and subsequently the RF port of the mixer for a homodyne measurement. This setup effectively separates the reflected microwave signal from the incident

signal, enabling the measurement of the quadrature part of the reflection coefficient by appropriate setting of the LO phase shifter. The quadrature signal is subsequently digitized using a sampling rate of 200 kS/s. Power spectral density in Fig. 3 and Fig. 4 is calculated with Welch's power spectral density estimate. For power measurements (Fig 1C), an IQ mixer is used to measure both field quadratures, and the total power is then computed. See Fig. S1 for details.

The diamonds (3 mm \times 3 mm \times 0.9 mm, 4 ppm NV ensemble, sourced from Element 6) are set at the TE01 δ mode maximum point in the center of the dielectric resonator (Skyworks, $\epsilon_r \approx 31$). A wafer of 4H-SiC is used for heat transfer and supporting the diamond, while two pieces of low-loss-tangent polytetrafluoroethylene (PTFE) are used to fix and align the dielectric resonator. An aluminum shield is employed to isolate the system from external signals in the lab, such as WiFi and 3G signals (1.9 GHz), and to reduce radiative losses. An 8W 532 nm pump laser is utilized to optically polarize the spin ensemble. External magnetic bias is provided by 3-axis magnetic coils.

In the noise measurements, we first replace the cavity with a 50 Ω resistor to set a baseline for the Johnson-Nyquist limit. Then we put back the cavity and gradually changed the tunable attenuator to test the spin refrigeration with different input microwave power.

- system description: uniform coupling between each NV and the cavity mode is a valid assumption only if the mode peak area \gg NV ensemble size. cavity details not given so cannot judge the validity of this assumption.

We thank the reviewer for pointing this out. We added simulation results for the cavity mode in the supplementary materials to show that the mode peak area much larger than NV ensemble size.

Change in the text: We add a figure for the cavity mode profile in the Supplementary Materials to show that the mode peak area \gg NV ensemble size. We also add a paragraph to describe this in the Supplementary Materials:

The diamonds (3 mm \times 3 mm \times 0.9 mm, 4 ppm NV ensemble, sourced from Element 6) are set at the TE01 δ mode maximum point in the center of the dielectric resonator (Skyworks, $\epsilon_r \sim 31$). See Fig. S1(b) and S1(c) for cavity mode simulation and reflection measurements. Note that the diamond size is much smaller than the mode homogeneous region so it is fair to assume a homogeneous single spin coupling strength g_s .

- figure 1C and related text: what is Δ_s ? is it related to the Δ_j in the main text (line 105)?

Δ_s is the spin-cavity detuning (i.e. the detuning between the ensemble line center and the cavity) while Δ_j is the cavity-probe detuning for each spin j .

Change to text: we define ω_j and ω_s in the second paragraph of main text Sec. IIA. The spin detuning, Δ_s , is defined in the caption of main text Fig. 1. Relevant definitions are listed here:

$\Delta_j = \omega_j - \omega_d$ is the detuning between the NV transition frequency ω_j and the drive frequency ω_d , and $\Delta = \omega_c - \omega_d$ is the detuning between the cavity frequency ω_c and the drive frequency.

We model the $N \gg 1$ NV centers as non-interacting two-level systems with transition frequencies ω_j , distributed inhomogeneously due to heterogeneous local magnetic and strain environments as well as hyperfine coupling with ^{14}N nuclear spins (Fig. 1b). The center frequency of the spin transitions, ω_s , is tuned by modulating the amplitude of a uniform magnetic field along the diamond's [100] axis.

c, Reflection $|r|^2$ with cavity detuning $\Delta = \omega_c - \omega_d$ and $\Delta_s = \omega_c - \omega_s$.

- more on figure 1C: the choice of parameters is not trivial for non-expert readers: it must be stated that the tuning of the NV resonances relative to cavity resonance is done by an external magnetic field which is applied like is done in ODMR experiments, I did not find this in the text, and it should be drawn in figure 1.

We thank the reviewer for pointing this out. We made Fig. 1b to be more clear and explicitly indicate the bias magnetic field and how it tunes the NV resonances relative to cavity resonance.

Change in the text: We redraw the Fig. 1b, and add a sentence in the caption: An external magnetic field is applied to tune the NV transitions to be on resonance with the cavity frequency. We also draw an external magnetic field in Fig. 1b to make it to be more clear.

- figures 1C and 2A disagree on the amount of reflection $|r|^2$ for zero detuning: figure 1C suggests near zero reflection while figure 2A suggests reflection of $\approx 20\%$ (0.2). which is it ? does the cavity really offer near 0 reflection on it's resonant frequency ? what is the true reflection when on NV resonance ? is figure 1C normalized ?

We thank the reviewer for the question. Actually, the figures 1C and 2A agree with each other. Figure 2A is a plot of the value of the point ($\Delta = 0$, $\Delta_s = 0$) as microwave power is varied. This point is centered in figure 1C and is purple rather than black, and the value is 42% with a microwave power input of -52 dBm. This is because we have polariton splitting of the cavity mode when the spin ensemble is on resonance, changing the cavity reflection from near-zero (critical coupling) to a reflective state. Figure 1C is normalized by the reflection signal with detuning $\Delta = 5$ MHz, where the cavity-probe detuning is large enough so that we can assume the reflection power is the same as the input power.

- figures 1C and 2d use opposite color coding which I find confusing.

We thank the reviewer for the suggestion. We want to keep most of the regions white and the regions we want to highlight with black, but some quantities we seek to minimize and others are maximized. This will highlight the points we want to emphasize and a consistent coloring throughout the manuscript, but if there are strong feelings on this point or requirements/suggestions on the figure color coding formatting we are open to changing the colormap.

- line 49-410 : claims of applications in time-keeping, etc seem pre-mature, this experiment is very far from state-of-the-art in e.g. atomic clocks in both accuracy and integrability.

We thank the reviewer for pointing this out. We agree that this experiment is very far from the state-of-the-art in those directions, but we want to point out that NV centers are not, in principle, limited to magnetometry only. We have removed references to other applications in the abstract and introduction where they could come across as proximal, and only kept a sentence in the outlook section.

Change in the text: We removed references to other applications in the abstract and introduction, and only kept a sentence in the outlook section.

-
- [1] I. Fescenko, A. Jarmola, I. Savukov, P. Kehayias, J. Smits, J. Damron, N. Ristoff, N. Mosavian, and V. M. Acosta, Diamond magnetometer enhanced by ferrite flux concentrators, *Phys. Rev. Res.* **2**, 023394 (2020).
- [2] E. R. Eisenach, J. F. Barry, M. F. O’Keeffe, J. M. Schloss, M. H. Steinecker, D. R. Englund, and D. A. Braje, Cavity-enhanced microwave readout of a solid-state spin sensor, *Nature Communications* **12**, 1357 (2021).
- [3] R. Wilcox, E. Eisenach, J. Barry, M. Steinecker, M. O’Keeffe, D. Englund, and D. Braje, Thermally polarized solid-state spin sensor, *Phys. Rev. Appl.* **17**, 044004 (2022).
- [4] G.-Q. Zhang, Z. Chen, D. Xu, N. Shammah, M. Liao, T.-F. Li, L. Tong, S.-Y. Zhu, F. Nori, and J. Q. You, Exceptional point and cross-relaxation effect in a hybrid quantum system, *PRX Quantum* **2**, 020307 (2021).
- [5] Y. Zhang, Q. Wu, S.-L. Su, Q. Lou, C. Shan, and K. Mølmer, Cavity quantum electrodynamics effects with nitrogen vacancy center spins coupled to room temperature microwave resonators, *Physical Review Letters* **128**, 253601 (2022).
- [6] J. F. Barry, J. M. Schloss, E. Bauch, M. J. Turner, C. A. Hart, L. M. Pham, and R. L. Walsworth, Sensitivity optimization for nv-diamond magnetometry, *Rev. Mod. Phys.* **92**, 015004 (2020).
- [7] A. Gupta, L. Hacquebard, and L. Childress, Efficient signal processing for time-resolved fluorescence detection of nitrogen-vacancy spins in diamond, *J. Opt. Soc. Am. B* **33**, B28 (2016).
- [8] J. F. Barry, M. J. Turner, J. M. Schloss, D. R. Glenn, Y. Song, M. D. Lukin, H. Park, and R. L. Walsworth, Optical magnetic detection of single-neuron action potentials using quantum defects in diamond, *Proceedings of the National Academy of Sciences* **113**, 14133 (2016).
- [9] J. M. Schloss, J. F. Barry, M. J. Turner, and R. L. Walsworth, Simultaneous broadband vector magnetometry using solid-state spins, *Phys. Rev. Appl.* **10**, 034044 (2018).
- [10] I. Fescenko, A. Jarmola, I. Savukov, P. Kehayias, J. Smits, J. Damron, N. Ristoff, N. Mosavian, and V. M. Acosta, Diamond magnetometer enhanced by ferrite flux concentrators, *Phys. Rev. Res.* **2**, 023394 (2020).
- [11] J. F. Barry, M. H. Steinecker, S. T. Alsid, J. Majumder, L. M. Pham, M. F. O’Keefe, and D. A. Braje, Sensitive ac and dc magnetometry with nitrogen-vacancy center ensembles in diamond, arXiv preprint arXiv:2305.06269 (2023).
- [12] G. Chatzidrosos, A. Wickenbrock, L. Bougas, N. Leefer, T. Wu, K. Jensen, Y. Dumeige, and D. Budker, Miniature cavity-enhanced diamond magnetometer, *Phys. Rev. Appl.* **8**, 044019 (2017).
- [13] D. Le Sage, L. M. Pham, N. Bar-Gill, C. Belthangady, M. D. Lukin, A. Yacoby, and R. L. Walsworth, Efficient photon detection from color centers in a diamond optical waveguide, *Phys. Rev. B* **85**, 121202 (2012).
- [14] T. Wolf, P. Neumann, K. Nakamura, H. Sumiya, T. Ohshima, J. Isoya, and J. Wrachtrup, Subpicotesla diamond magnetometry, *Phys. Rev. X* **5**, 041001 (2015).
- [15] C. L. Degen, F. Reinhard, and P. Cappellaro, Quantum sensing, *Rev. Mod. Phys.* **89**, 035002 (2017).
- [16] H. Clevenson, M. E. Trusheim, C. Teale, T. Schröder, D. Braje, and D. Englund, Broadband magnetometry and temper-

- ature sensing with a light-trapping diamond waveguide, *Nature Physics* **11**, 393 (2015).
- [17] M. W. Doherty, N. B. Manson, P. Delaney, F. Jelezko, J. Wrachtrup, and L. C. Hollenberg, The nitrogen-vacancy colour centre in diamond, *Physics Reports* **528**, 1 (2013), the nitrogen-vacancy colour centre in diamond.
- [18] J. Rovny, Z. Yuan, M. Fitzpatrick, A. I. Abdalla, L. Futamura, C. Fox, M. C. Cambria, S. Kolkowitz, and N. P. de Leon, Nanoscale covariance magnetometry with diamond quantum sensors, *Science* **378**, 1301 (2022).
- [19] C. Du, T. van der Sar, T. X. Zhou, P. Upadhyaya, F. Casola, H. Zhang, M. C. Onbasli, C. A. Ross, R. L. Walsworth, Y. Tserkovnyak, and A. Yacoby, Control and local measurement of the spin chemical potential in a magnetic insulator, *Science* **357**, 195 (2017).
- [20] S. Li, M. Huang, H. Lu, N. J. McLaughlin, Y. Xiao, J. Zhou, E. E. Fullerton, H. Chen, H. Wang, and C. R. Du, Nanoscale magnetic domains in polycrystalline mn_3sn films imaged by a scanning single-spin magnetometer, *Nano Letters* **23**, 5326 (2023), pMID: 37219013.
- [21] H. Wang, S. Zhang, N. J. McLaughlin, B. Flebus, M. Huang, Y. Xiao, C. Liu, M. Wu, E. E. Fullerton, Y. Tserkovnyak, and C. R. Du, Noninvasive measurements of spin transport properties of an antiferromagnetic insulator, *Science Advances* **8**, eabg8562 (2022).
- [22] M. Pelliccione, A. Jenkins, P. Ovarthaiyapong, C. Reetz, E. Emmanouilidou, N. Ni, and A. C. Bleszynski Jayich, Scanned probe imaging of nanoscale magnetism at cryogenic temperatures with a single-spin quantum sensor, *Nature Nanotechnology* **11**, 700 (2016).
- [23] E. Bauch, C. A. Hart, J. M. Schloss, M. J. Turner, J. F. Barry, P. Kehayias, S. Singh, and R. L. Walsworth, Ultralong dephasing times in solid-state spin ensembles via quantum control, *Phys. Rev. X* **8**, 031025 (2018).
- [24] P. L. Stanwix, L. M. Pham, J. R. Maze, D. Le Sage, T. K. Yeung, P. Cappellaro, P. R. Hemmer, A. Yacoby, M. D. Lukin, and R. L. Walsworth, Coherence of nitrogen-vacancy electronic spin ensembles in diamond, *Phys. Rev. B* **82**, 201201 (2010).
- [25] J. M. Taylor, P. Cappellaro, L. Childress, L. Jiang, D. Budker, P. R. Hemmer, A. Yacoby, R. Walsworth, and M. D. Lukin, High-sensitivity diamond magnetometer with nanoscale resolution, *Nature Physics* **4**, 810 (2008).
- [26] F. Dolde, H. Fedder, M. W. Doherty, T. Nöbauer, F. Rempp, G. Balasubramanian, T. Wolf, F. Reinhard, L. C. L. Hollenberg, F. Jelezko, and J. Wrachtrup, Electric-field sensing using single diamond spins, *Nature Physics* **7**, 459 (2011).
- [27] X. Wang, Y. Xiao, C. Liu, E. Lee-Wong, N. J. McLaughlin, H. Wang, M. Wu, H. Wang, E. E. Fullerton, and C. R. Du, Electrical control of coherent spin rotation of a single-spin qubit, *npj Quantum Information* **6**, 78 (2020).
- [28] H. Wang, M. E. Trusheim, L. Kim, H. Raniwala, and D. R. Englund, Field programmable spin arrays for scalable quantum repeaters, *Nature Communications* **14**, 704 (2023).
- [29] A. Jarmola, S. Lourette, V. M. Acosta, A. G. Birdwell, P. Blümler, D. Budker, T. Ivanov, and V. S. Malinovsky, Demonstration of diamond nuclear spin gyroscope, *Science Advances* **7**, eabl3840 (2021), <https://www.science.org/doi/pdf/10.1126/sciadv.abl3840>.
- [30] V. V. Soshenko, S. V. Bolshedvorskii, O. Rubinas, V. N. Sorokin, A. N. Smolyaninov, V. V. Vorobyov, and A. V. Akimov, Nuclear spin gyroscope based on the nitrogen vacancy center in diamond, *Phys. Rev. Lett.* **126**, 197702 (2021).
- [31] K.-M. C. Fu, G. Z. Iwata, A. Wickenbrock, and D. Budker, Sensitive magnetometry in challenging environments, *AVS Quantum Science* **2** (2020).
- [32] Y. Anahory, J. Reiner, L. Embon, D. Halbertal, A. Yakovenko, Y. Myasoedov, M. L. Rappaport, M. E. Huber, and E. Zeldov, Three-junction squid-on-tip with tunable in-plane and out-of-plane magnetic field sensitivity, *Nano letters* **14**, 6481 (2014).
- [33] E. Trabaldo, C. Pfeiffer, E. Andersson, M. Chukharkin, R. Arpaia, D. Montemurro, A. Kalaboukhov, D. Winkler, F. Lombardi, and T. Bauch, Squid magnetometer based on grooved dayem nanobridges and a flux transformer, *IEEE Transactions on Applied Superconductivity* **30**, 1 (2020).

- [34] W. C. Griffith, S. Knappe, and J. Kitching, Femtotesla atomic magnetometry in a microfabricated vapor cell, *Optics express* **18**, 27167 (2010).
- [35] C. Bonato, M. S. Blok, H. T. Dinani, D. W. Berry, M. L. Markham, D. J. Twitchen, and R. Hanson, Optimized quantum sensing with a single electron spin using real-time adaptive measurements, *Nature nanotechnology* **11**, 247 (2016).
- [36] D. P. Fahey, K. Jacobs, M. J. Turner, H. Choi, J. E. Hoffman, D. Englund, and M. E. Trusheim, Steady-state microwave mode cooling with a diamond n-v ensemble, *Phys. Rev. Appl.* **20**, 014033 (2023).
- [37] H. Wu, S. Mirkhanov, W. Ng, and M. Oxborrow, Bench-top cooling of a microwave mode using an optically pumped spin refrigerator, *Physical Review Letters* **127**, 053604 (2021).
- [38] Y. Zhang, Q. Wu, H. Wu, X. Yang, S.-L. Su, C. Shan, and K. Mølmer, Microwave mode cooling and cavity quantum electrodynamics effects at room temperature with optically cooled nitrogen-vacancy center spins, *npj Quantum Information* **8**, 125 (2022).
- [39] W. Ng, H. Wu, and M. Oxborrow, Quasi-continuous cooling of a microwave mode on a benchtop using hyperpolarized nv-diamond, *Applied Physics Letters* **119** (2021).
- [40] E. Bauch, S. Singh, J. Lee, C. A. Hart, J. M. Schloss, M. J. Turner, J. F. Barry, L. M. Pham, N. Bar-Gill, S. F. Yelin, and R. L. Walsworth, Decoherence of ensembles of nitrogen-vacancy centers in diamond, *Phys. Rev. B* **102**, 134210 (2020).
- [41] D. H. Meyer, J. C. Hill, P. D. Kunz, and K. C. Cox, Simultaneous multiband demodulation using a rydberg atomic sensor, *Physical Review Applied* **19**, 014025 (2023).
- [42] H. Mamin, M. Kim, M. Sherwood, C. T. Rettner, K. Ohno, D. Awschalom, and D. Rugar, Nanoscale nuclear magnetic resonance with a nitrogen-vacancy spin sensor, *Science* **339**, 557 (2013).
- [43] I. Dutta, D. Savoie, B. Fang, B. Venon, C. Garrido Alzar, R. Geiger, and A. Landragin, Continuous cold-atom inertial sensor with 1 nrad/sec rotation stability, *Physical review letters* **116**, 183003 (2016).

Reviewer 1 (Remarks to the Author):

In the current work, the authors had not only made a great effort to optimize the microwave-based readout scheme proposed in 2021 [Nat. Commun 12:1357 (2021)], but also provided a comprehensive study on the response of the NV center spins-microwave cavity system to the microwave power and laser illumination. In particular, the authors have demonstrated for the first time that the microwave mode cooling, as revealed by several studies in recent years, can be utilized to increase sensitivity of magnetometer. Although not fully exposed in the main text, the theoretical treatment of the inhomogeneous broadening and the strong driving is not trivial at all. What more important is that the authors have made a great effort to achieve the agreement between the experiment and the theory. Based on these reasons, I recommend the publication of the paper.

In comparison to the last version of the manuscript, the authors have made significant efforts to address my and other reviewers' comments and to improve the readability. I am satisfied with all their responses. By reading through the revised manuscript and supporting information, I have also spotted some minor problems, and indicate them below.

We thank the reviewer once again for their positive comments and recommendation of our paper. The reviewer's suggestions have been invaluable in significantly enhancing the manuscript. We prepare a point-by-point answer with change in the text for the comments below.

Comments for the main text:

Comment 1: It seems to me that the authors have emphasized the spin-refrigeration more than the non-linear model in the title and introduction. However, in the text, the authors have explained first the non-linear model, and then addressed the spin-refrigeration. It seems to me that the spin-refrigeration is also part of the results from the non-linear model. Thus, I suggest the authors to revise some parts of the title and introduction to reflect the relation between the non-linear model and the spin-refrigeration effect.

We thank the reviewer for the suggestion. We had another round of discussion about changing the title and we find the title now is concise and it is hard to add several words to discuss nonlinearity in the title. However, we emphasize the interplay between the nonlinear model and spin refrigeration in the introduction part as the reviewer suggests.

Change in the text: In this work, we address both of these limitations. We comprehensively model nonlinear spin saturation effects in the strong coupling regime, including spin ensemble inhomogeneity and optical polarization dynamics. We further apply recently introduced "spin refrigeration" [1-4] techniques to an actively-probed sensor for the first time, reducing the noise floor below the Johnson-Nyquist limit. We combine these two effects in the nonlinear regime to optimize device performance, achieving a sensitivity of $576 \pm 6 \text{ fT}/\sqrt{\text{Hz}}$ with an inverse readout fidelity of $\sigma_e \sim 360$ which sets the state-of-the-art for a continuous-wave NV magnetometer.

Comment 2: It seems that there are too many symbols in the caption of Fig. 1, and they might be overwhelming to the readers. There are some symbols not explained, for example α , ω_c , ω_d , ω_s , β . Some of the symbols such as Γ , γ do not appear in the figure. The caption might not be the right place to introduce them. The energy diagram shown in Fig. 1(b) is too simplified, and the indication of the level transitions is not correct. It might also be necessary to

explain what is the strong coupling features.

We thank the reviewer for the suggestions. Although there are a number of symbols in the figure, we would like to show some physical intuition for what the key parameters of the system are (e.g. cavity loss rates, spin coupling) by indicating them on a schematic image. We have now properly described what each symbol indicates in the figure caption. We have also redrawn the energy diagram to more clearly explain the parameter relationships. We also add a sentence in the caption to discuss the avoid-crossing features.

Change in the text: We redraw the Fig. 1b and change the caption.

Comment 3: In the main text, it might be necessary to specify also the magnitude of the applied magnetic field, and the frequency of the cavity and the spin transitions. This information is necessary if someone wants to reproduce the results in the paper.

We thank the reviewer for pointing this out. We add information about the cavity frequency, applied magnetic field, etc, in the Methods section. There is also a cavity mode simulation and experiment measurement for the reflection in the Supplementary Information Fig. S1.

Change in the text: The diamonds ($3 \text{ mm} \times 3 \text{ mm} \times 0.9 \text{ mm}$, 4 ppm NV ensemble, sourced from Element 6) are set at the TE_{01δ} mode maximum point in the center of the dielectric resonator (Skyworks, $\epsilon_r \sim 31$, center frequency: 2.877 GHz). A wafer of 4H-SiC is used for heat transfer and supporting the diamond, while two pieces of low-loss-tangent polytetrafluoroethylene (PTFE) are used to fix and align the dielectric resonator. An aluminum shield is employed to isolate the system from external signals in the lab, such as WiFi and 3G signals (1.9 GHz), and to reduce radiative losses. An 8W 532 nm pump laser is utilized to optically polarize the spin ensemble. External magnetic bias is provided by 3-axis magnetic coils. Magnetic field along [100] direction has an amplitude of $\sim 4.3 \text{ G}$ to move the NV resonant frequency to be on resonance with the cavity frequency.

Comment 4. The author might need to explain where the factor $\sqrt{3}$ in Eq. (1) comes from.

Change in the text: We add a sentence to describe the origin of $\sqrt{3}$.

Comment 5: In Fig. 2(a,c,d), the symbol $\chi = \sqrt{2}$ is indicated besides the dash-dotted lines. However, in the caption of Fig. 2, there is no explanation of this expression and the dash-dotted lines.

We thank the reviewer for spotting this. We add a sentence to describe the saturation power P_s in the figure caption.

Change in the text: We add a sentence to describe the saturation power P_s in the figure caption.

Comment 6: By looking carefully Fig. 3(b), one can see that the optimal sensitivity is achieved by a microwave

power around -18 dBm rather than -22 dBm. The optimal microwave power -18 dBm is consistent with the power to achieve the largest signal, see Fig. 2c. By examining carefully Fig. 3a, it seems that the microwave noise starts increasing due to heating for the microwave power larger than around -22 dBm. However, the spin refrigeration compensates partly the heating, and improves the sensitivity. Thus, the power -22dBm can only make sense as the power to avoid the increased noise due to the microwave heating.

By examining Fig. 3b, one can see that the difference between the J-N-limit and the achieved sensitivity is large for weak microwave power, but becomes almost negligible for strong microwave power. At the optimal microwave power -18 dBm, the difference is almost nothing. This suggests that the sensitivity enhancement due to the spin refrigeration effect, is more significant at low microwave power.

The above conclusion can be much clearly seen in the plot of ratio $\Delta\eta/\eta_0$ of the sensitivity reduction $\Delta\eta$ to the sensitivity η_0 . I believe that η_0 is the sensitivity when the system operates in the magnetic insensitive mode. At the microwave power -18 dBm, this ratio is zero, indicating no enhancement. At the microwave power -22 dBm, this ratio is 0.51 dB, indicating a sensitivity enhancement. For lower microwave power, the ratio is about -1dB, indicating a much larger enhancement

From the above discussion, we can see that there is a conflict between the achieved optimal sensitivity and the spin refrigeration-induced enhancement. It would be helpful if the authors could explain clearly under which condition the spin refrigeration effect can enhance the sensitivity, and why they choose -22 dBm as the operating point instead of -18 dBm.

We thank the reviewer for pointing this out. We first want to clarify that the cooling can benefit with all the powers we used in the experiment. The reason why we see total noise is larger than thermal noise is due to the phase noise, not the heating induced by the increased power. In Fig. 3a blue curve, we measure the total noise (phase noise + thermal noise) and the red curve shows the noise with cooling (phase noise + cooled thermal noise). We can see that the blue curve is always larger than the red curve, meaning that cooling can always enhance the sensitivity. The green curve in Fig. 3a shows the difference between the red and blue curves, related to the cooling-induced sensitivity enhancement. We add more details about this process in the main text.

After considering the cooling effect, the sensitivity floor is unchanged within our measurement error between -18 dBm and -22 dBm (centered at -20.3 dBm). We choose -22 dBm for the PSD measurement (frequency-resolved sensitivity) as the low-frequency flicker noise and high-frequency phase noise contributions are lower at lower oscillator power; these do not affect the optimal sensitivity, which is in the thermal-limited frequency band, but do affect the overall noise spectrum.

Change in the text: This spin refrigeration effect begins to be suppressed at the same saturation onset power $P_s = -35$ dBm described in Eq. (5) as the effective coupling strength is diminished, with a reduction by $1/e$ at -23 dBm. Oscillator phase noise also contributes to the overall noise performance in this region, increasing both the cooled (red) and un-cooled (blue) noise floors. At the sensitivity-maximizing operating point $P = -22$ dBm (note that this is different from the signal optimal point -18 dBm), we observe a noise power reduction of 0.51 dB, demonstrating that spin refrigeration offers sensitivity enhancement for an actively-driven system.

Change in the text: Next we examine the performance of the device in the frequency domain. We operate at a fixed power of -22 dBm which produces near-optimal sensitivity in the thermal-limited regime as described above, while also reducing the contributions of power-dependent flicker and phase noises (see Supplemental Note III).

Comment 7: In the equations (2) and (5), there is a period in the end. However, in the equations (5) and (6), the period is missed. There are also similar problems for the equations in the supporting information.

We thank the reviewer for pointing this out. We have added period and coma for all equations.

Change in the text: We have added punctuation for all equations.

Comment 8: In the second paragraph of Section II B, the text said “To establish a noise baseline, we detune the spin ensemble from the cavity by 5 MHz and measure the noise power spectral density in a 1 Hz band at 15 kHz offset in the signal channel for the bare cavity driven on resonance (blue).”. I suggest the authors to add one sentence to relate this description to Fig. 4, and explain why they have chosen 5 kHz offset.

We thank the reviewer for pointing this out. We add a description when talking about the noise behavior.

Change in the text: As we will show in Fig. 4, the noise floor at 15 kHz is dominated by the thermal noise but without the flicker noise components.

Comment 9: In Fig. 5(b), there is a solid line to distinguish between the normal region and the bistable region. It might be necessary to explain shortly how the biostability occurs and what effect it might have on the sensing performance.

We thank the reviewer for pointing this out. We add a figure in the bistability discussion in the Supplementary Materials.

We add a sentence in main text and add a figure in supplement bistability section.

Comment 10: In the second paragraph of Section IV, the authors have discussed about low- ρ , medium- ρ and high- ρ regime. It might be necessary to give some numbers, for example, low- ρ regime ($\rho < \text{ppm}$). It might be useful to introduce some vertical lines in Fig. 5(a). There is also a typo for the left axis of Fig. 5(a), and the correct symbol should be V_d/V .

We thank the reviewer for pointing this out. We have added the numbers related to the NV density. We also fixed the typo the reviewer mentioned.

Change in the text:

In the low- ρ regime ($\rho < 0.02$ ppm), laser-induced dephasing dominates both the inhomogeneous and homogeneous NV linewidth; therefore, adding more density to the system increases the collective coupling without the cost of broadening the NV spins. In the medium- ρ regime ($0.02 \text{ ppm} < \rho < 3 \text{ ppm}$), the density-induced inhomogeneous broadening eclipses the laser-induced dephasing, and the sensitivity is almost independent of density ρ . In the high- ρ regime ($\rho > 3 \text{ ppm}$), the spin dipole-dipole homogeneous dephasing starts dominating the laser-induced dephasing, causing a sensitivity drop in this domain.

Comment 11: In Section XI B, the authors indicate that they have used their device to measure a DC field around 4 uT. However, in section IVC of the supporting information, the authors have measured an AC field with amplitude 2 nT. The authors might need to mention the latter measurement in the main text. Besides, from Fig. 4 and Fig. S9(b), we can see that the sensitivity for low frequency is around nT. Would it possible to detect such a small field by averaging the oscillating noise over longer time?

We thank the reviewer for the comment. We add an AC test magnetic field discussion in the Method section. The relatively worse sensitivity in the low frequency is not due to the sensor itself. It is due to the environment's magnetic field (earth field, for example). We have several pieces of evidence on this point: (1). In the Supplementary Materials, we measure the responsivity dependence with different frequencies. The result shows that the responsivity is flat over different frequencies. (2). We turn off the spin by turning off the laser and measure the noise floor when the magnetic field sensitivity is off. The results are shown in the green line in Fig. 4. It shows an effective sensitivity given the responsivity is flat (as we just explained in (1)) ($2 \text{ pT}/\sqrt{\text{Hz}}$) around 10 Hz. In this process, we neglect the possible low-frequency noise intrinsic to the diamond. We described this whole process in line 325-338 of the main text.

Change in the text: An AC test magnetic field is also shown in the Supplementary Information.

Comments for the supporting information:

Comment 1: Add common or period to the end of expressions when necessary. Mark the equations as “(S1),(S2),..”, the figures as “Supplementary Figure S1,...”.

We thank the reviewer for the suggestion. We made changes based on reviewer's comments and author guidance file from Nature Communications. Specifically, we marked Supplementary Figures as "Supplementary Figure S1" and equations as Supplementary Eq. (1), etc.

Comment 2: In the main text and the supporting information, the symbol $|\alpha|^2$ is interpreted as the cavity occupancy. I am not familiar with the term “cavity occupancy”. It seems to me that $|\alpha|^2$ represents the intra-cavity photon number. The authors might need to check this term.

Both cavity occupancy and intra-cavity photon number are used for $|\alpha|^2$. We use these two terms back and forth

to avoid the repeated use of the terms.

Comment 3: It might be necessary to illustrate the bistability with a picture, and explain how it might affect the sensitivity.

We thank the reviewer for pointing this out. We add a figure in the bistability discussion in the Supplementary Materials.

We add a sentence in main text and add a figure in supplement bistability section.

Change in the text: We plot the multiple solutions in Supplementary Information Fig. 6. Within the critical power, there are more than one steady-state solution [5]. There are two stable states (yellow and blue) and an unstable solution (red). Operating on the unsaturated branch will give a better sensitivity based on this modeling, but optimal performance requires operation close to the edge of the bistable phase and potentially results in a nonlinear magnetic response. A full treatment of magnetometer operation deep in the bistable regime is needed to explore these possibilities.

Change in the text: We add Supplementary Information Fig. 6 to describe this.

Comment 4: The theoretical treatment in Section II D is very similar to what did in the paper [npj Quantum Information 8, 125 (2022)], see the Supplementary Note 2 of that paper. It might be necessary to cite this paper.

We thank the reviewer for pointing this out. We have added the citation for this paper in the Supplementary Information.

Change in the text: We add the citation of the paper mentioned by the reviewer.

Comment 5: What does the value 1.5×10^7 mean in the end of the section IVA?

It shows the ratio between the maximum detectable field in the linear regime (dynamical range) and the sensitivity at one second.

Change in the text: The ratio between the dynamical range and the minimum detectable field with 1 Hz measurement bandwidth is 1.5×10^7 .

Reviewer 2 (Remarks to the Author):

The authors have almost entirely re-written this manuscript in response to both my, and the other reviewers comments. My first response is that the manuscript reads significantly better than the first revision and is much easier to follow. The responses to my comments are very comprehensive with clear changes evident in the manuscript.

I also appreciate the authors emphasizing the novelty of the work specific to this manuscript and I find myself

significantly more onboard than I was after reading the first version.

Overall a comprehensive response to my comments and a significant alteration of the manuscript (for the better). I am afraid I have no more useful comments for the authors, and have no issues recommending publication.

We thank the reviewer again for the helpful suggestions, and we also feel the comments from the reviewer helped improve the manuscript significantly.

Reviewer 3 (Remarks to the Author)

the Authors have addressed and answered all the comments or questions I have raised in the first review. I have no further issues. I remain in the opinion that this manuscript will fit better in a specialized physics publication, but I find the scientific work and presentation ready for publication.

We thank the reviewer once again for the valuable suggestions, which we believe have greatly enhanced the quality of the manuscript.

-
- [1] D. P. Fahey, K. Jacobs, M. J. Turner, H. Choi, J. E. Hoffman, D. Englund, and M. E. Trusheim, Steady-state microwave mode cooling with a diamond $n-v$ ensemble, *Phys. Rev. Appl.* **20**, 014033 (2023).
 - [2] H. Wu, S. Mirkhanov, W. Ng, and M. Oxborrow, Bench-top cooling of a microwave mode using an optically pumped spin refrigerator, *Physical Review Letters* **127**, 053604 (2021).
 - [3] Y. Zhang, Q. Wu, H. Wu, X. Yang, S.-L. Su, C. Shan, and K. Mølmer, Microwave mode cooling and cavity quantum electrodynamics effects at room temperature with optically cooled nitrogen-vacancy center spins, *npj Quantum Information* **8**, 125 (2022).
 - [4] W. Ng, H. Wu, and M. Oxborrow, Quasi-continuous cooling of a microwave mode on a benchtop using hyperpolarized $n-v$ diamond, *Applied Physics Letters* **119** (2021).
 - [5] A. Angerer, S. Putz, D. O. Krimer, T. Astner, M. Zens, R. Glattauer, K. Streltsov, W. J. Munro, K. Nemoto, S. Rotter, J. Schmiedmayer, and J. Majer, Ultralong relaxation times in bistable hybrid quantum systems, *Science Advances* **3**, e1701626 (2017).